# Genetically engineered insects with sex-selection and genetic incompatibility enable population suppression

**Ambuj Upadhyay[1,2], Nathan R Feltman[1,2], Adam Sychla[1,2], Anna Janzen[1,2], Siba R Das[1,2], Maciej Maselko[1,2], Michael Smanski[1,2]***

[1]Department of Biochemistry, Molecular Biology, and Biophysics, University of Minnesota, Saint Paul, United States; [2]Biotechnology Institute, University of Minnesota, Saint Paul, United States

**Abstract** Engineered Genetic Incompatibility (EGI) is a method to create species-like barriers to sexual reproduction. It has applications in pest control that mimic Sterile Insect Technique when only EGI males are released. This can be facilitated by introducing conditional female-lethality to EGI strains to generate a sex-sorting incompatible male system (SSIMS). Here, we demonstrate a proof of concept by combining tetracycline-controlled female lethality constructs with a *pyramus*-targeting EGI line in the model insect *Drosophila melanogaster*. We show that both functions (incompatibility and sex-sorting) are robustly maintained in the SSIMS line and that this approach is effective for population suppression in cage experiments. Further we show that SSIMS males remain competitive with wild-type males for reproduction with wild-type females, including at the level of sperm competition.

## Editor's evaluation

This paper is of interest to entomologists caring for genetic pest control or molecular biologists following synthetic biology. The authors describe a fruit fly strain that combines constructs that establish both repressible female-lethality and genetic incompatibility based on CRISPR transactivation. They show that this strain has high penetrance for these two traits and that it can suppress wild-type flies when released into cycling cage populations.

**\*For correspondence:**
smanski@umn.edu

## Introduction

Arguably the most successful large-scale insect control approach to date is Sterile Insect Technique (SIT) (*Dyck et al., 2005*; *Barsanti et al., 1979*). SIT uses irradiation at sub-lethal doses to sterilize mass-reared insects prior to their targeted environmental release. Such an approach was successfully applied to eradicate the New World Screwworm (*Cochliomyia hominivorax*) from North and Central America over a 50-year period from the mid 1950s to early 2000s (*Scott et al., 2017*). SIT has been successfully applied for the broad-scale control of tephritid fruit flies (e.g. *Ceratitis capitata*, *Zeugodacus cucurbitae*, *Bactrocera tryoni*, and *Anastrepha ludens*), onion maggots (*Delia antiqua*), tse-tse flies (*Glossina* spp.), and several coleoptera and lepidoptera species (*Dyck et al., 2005*).

Several factors limit the widespread adoption of SIT for more insect pests. Exposing insects to sterilizing doses of irradiation can impact their longevity and mating competitiveness (*Lance et al., 2000*). Sometimes the lethal dose of radiation is sufficiently close to the sterilizing dose of radiation such that achieving 100% sterilization means that the vast majority of the males will not survive (*Ricardo Machi et al., 2019*). Many batch irradiation approaches do not separate males from females prior to release,

and this can both decrease the effectiveness (*Rendón et al., 2004*), and lead to public health concerns when the female insect is a disease vector (*Benedict, 2021*).

Recently, biotechnology-enabled approaches have been developed that have the potential to complement or replace SIT programs, or allow SIT-like programs that target a wider range of species. Integrating an early acting female lethality system to New World Screwworm without impacting male fitness and fecundity has the potential to improve SIT effectiveness by several fold (*Concha et al., 2020*). Release of insects with dominant lethality (RIDL), has been successfully demonstrated to suppress target mosquito populations in field trials in the Caribbean islands and South America (*Harris et al., 2012*). Precision-guided SIT (pgSIT) utilizes expression of Cas9 nuclease to knock out female survival and male fertility genes prior to generate potential biocontrol agents (*Li et al., 2021*; *Kandul et al., 2019*).

Engineered Genetic Incompatibility (EGI) is a strategy for creating bi-directional (male-to-female and female-to-male) mating incompatibility that mirrors the behavior of post-zygotic speciation mechanisms. (*Maselko et al., 2020*; *Maselko et al., 2017*; *Buchman et al., 2021*; *Waters et al., 2018*). We created EGI lines by engineering flies to express a dCas9-based programmable transcription activator (PTA) that targets the promoter of a gene which is lethal with expressed too high or in the tissues. This lethal over- or ectopic-expression is avoided in the EGI line by making it homozygous for a mutation that prevents binding of the PTA. Because the EGI line is homozygous for both the PTA and the promoter resistance mutation, it is true-breeding and has normal fecundity. However, outcrossing with wild-type produces hybrid offspring that inherit one copy of the PTA and only one copy of the resistance allele. Because the PTA is haplosufficient for lethality (i.e. one copy is lethal) and the resistance allele is haploinsufficient (i.e. strains must be homozygous for the resistance allele to survive in the pressence of the PTA), all hybrid offspring are inviable. Importantly, multiple mutually incompatible EGI lines can be created that are incompatible with wild-type and with each other, providing a way to rationally engineer negatively correlated cross-resistance for the first time (*Maselko et al., 2020*).

As with other underdominant biocontrol methods, EGI could be used as a threshold-dependent gene drive (*Magori and Gould, 2006*; *Davis et al., 2001*; *Buchman et al., 2021*). Alternatively, it could be coupled to a conditional female-lethal construct, such as the tetracycline (Tet)-repressible positive feedback circuit (*Fu et al., 2007*; *Li et al., 2014*), to produce a strain that more closely replicates SIT biocontrol agents. Here, we demonstrate the successful engineering of such a system in *Drosophila melanogaster*. We quantify the behavior of individual genetic components, perform male mating competition assays, and test the ability of SSIMS flies to suppress a wild-type population in multi-generational laboratory cage trials.

## Results

### Generating and validating SSIMS lines

To generate the SSIMS line, we made a stock that contains two female lethal (FL) constructs (*Das et al., 2020*) on the X chromosome and an EGI construct on the third chromosome (*Figure 1*). The female lethal construct contains 21 copies of the tet operator upstream of the Hsp70 promoter (*Li et al., 2014*). An intron from the *C. hominivorax tra* gene that exhibits sex-specific splicing (alternative 5' splice sites) is present immediately downstream of the start codon and upstream of a coding DNA sequence for the tet transactivator protein (*Figure 1c*). In females, this construct is expected to produce the tTA transgene, while the alternative splicing in males is expected to produce mature mRNA with several stop codons upstream of the tTA CDS. In comparing the behavior of strains containing one- or two-copies of this construct on the X chromosome, the two-copy strains showed a stronger female lethal phenotype, so we used a two-copy strain to construct SSIMS flies (*Das et al., 2020*). The EGI construct comprised a dCas9-VPR programmable transcriptional activator (PTA) targeting the promoter of a critical developmental morphogen gene (*Figure 1d*). In this study, we use a previously described EGI genotype that leverages lethal ectopic expression of the *pyramus* gene for hybrid incompatibility, but other genotypes are expected to work equally well. Throughout the strain-construction process, a mutation in the promoter of the *pyramus* gene was maintained to prevent binding of the PTA to the *pyramus* promoter, and therefore lethal ectopic expression (*Appendix 1—figure 1*). The SSIMS line used the *foxo* promoter to drive the PTA, as this genotype provided strong genetic incompatibility in previous studies (*Maselko et al., 2020*). Following the

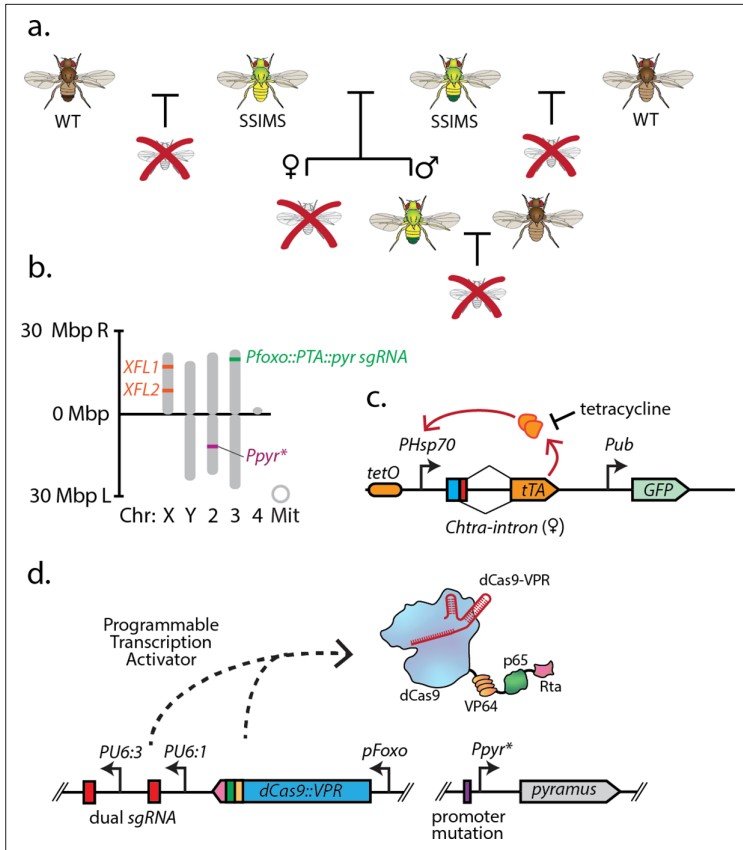

**Figure 1.** Overall design and implementation of Sex-sorting Incompatible Male System (SSIMS). (**a**) Illustration of desired behavior of SSIMS insects (green-yellow) when grown in the absence of Tet. Brown flies represent wild-type, and red crosses represent inviable offspring. (**b**) Genome-scale location of engineered loci in *D. melanogaster* SSIMS line. (**c,d**) Genetic cassette diagrams for the female lethal (FL) locus. The orange oval represents 21 repeats of the tet operator. The blue box represents the start codon. The red box denotes the exon sequence containing multiple stop codons that is retained when an alternative 5' splice site is utilized in males. Female splicing fuses the start codon to the tTA coding DNA sequence. (**d**) Genetic cassette diagram for the engineered genetic incompatibility (EGI) loci, respectively. SBOL iconography is used for genetic part representation. FL, X-linked female lethal construct; Mbp, megabasepairs; kb, kilobases; Chr, chromosome; Mit, mitochondrial; PTA, Programmable Transcriptional Activator.

mating scheme outlined in Supplementary Note 1, we were able to generate viable SSIMS flies, which contained both of the visual markers of the FL genotype (GFP expression) and the EGI genotype (red eyes) (*Figure 2a*). Other EGI genetic designs (*Maselko et al., 2020*; *Buchman et al., 2021*) or FL designs (*Yan and Scott, 2015*) are predicted to work equally well for engineering SSIMS.

## Efficacy of genetic components in SSIMS line

To validate the SSIMS line we first tested whether the two underlying behaviors (FL and EGI) were maintained. We confirmed that the SSIMS line bred true in the presence of Tet at 100 µg/ml and contained both visual markers for FL (GFP) and EGI (mini-white) (*Figure 2a*). In the presence of Tet at (10 µg/ml) the SSIMS line shows a slight sex-ratio bias with fewer females (40% female) surviving to adulthood than expected 50% (*Figure 2b*). This suggests that the FL transgene is partially leaky in females at 10 µg/ml Tet. In the absence of Tet no females survive to adulthood and either die as larvae or pupae (*Figure 2b*). While we did not empirically characterize sex-specific splicing, the phenotype of the SSIMS line suggests that sex-specific splicing occurs as previously reported for this genetic construct (*Li et al., 2014*). To confirm the behavior of the EGI components, 20 wild-type females were mated with either four wild-type males or four SSIMS males, then we quantified number of eggs laid per vial in 24 hr and total number of offspring produced per vial (24 hr egg-lay). Wild-type females

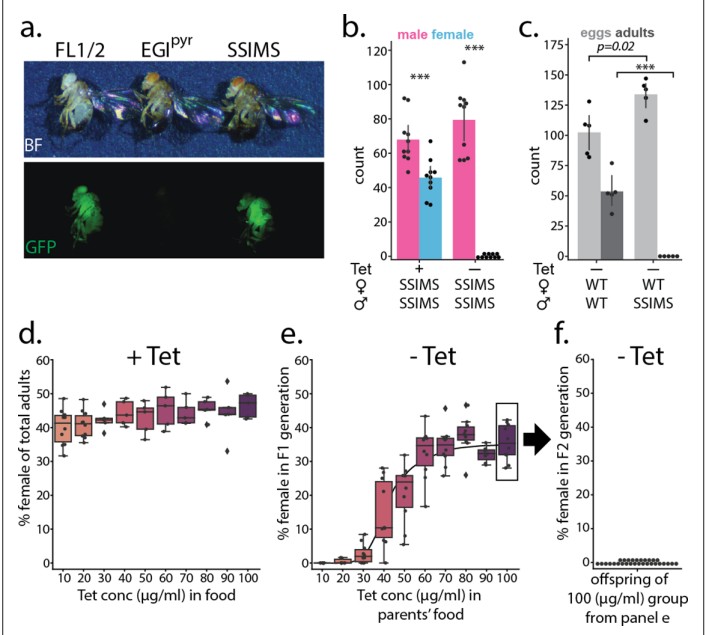

**Figure 2.** Phenotype of SSIMS flies. (**a**) Visual markers present in homozygous FL lines (GFP, left) and EGI lines (red eyes, center) are present in SSIMS flies generated via the mating scheme in **Appendix 1—figure 1**. (**b**) Counts of male (pink) and female (blue) offspring of SSIMS strain raised in food with 10 μg/ml Tetracyline (Tet) and without Tet. No females offspring were observed when larvae are raised in the absence of Tet. Chi-squared test shows significant difference from expected 50:50 ratio of males:female, n = 10 per group, *** denotes < 0.0001. (**c**) Counts of egg-lay (light gray) and surviving adults (dark gray) of wild-type females crossed with either wild-type or SSIMS males. SSIMS male mate successfully with wild-type females, producing slightly more eggs, but no adult offspring emerge, n = 5 per group, *** denotes p < 0.0001. (**d**) Percent female offspring when SSIMS stocks are reared in food with increasing concentrations of Tet, n = 10 per group. (**e**) Percent female offspring when respective progeny from (**d**) are reared in food without Tet, n = 7–10 per group. Line behind data shows Hill function that fits data with statistics described in the main text. (**f**) Percent female offspring when progeny from the 100 μg/ml group in (**e**) are reared in food without Tet, n = 30. FL, X-linked Female Lethal; EGI *pyr*, engineered genetic incompatibility targeting *pyramus*; WT, wild-type; SSIMS, sex-sorting incompatible male system; Tet, Tetracycline.

did not produce any viable adults when mated with SSIMS male (**Figure 2c**). There was no difference in fecundity of the wild-type and SSIMS strains, when mated to their own type, in a head-to-head comparison (**Appendix 1—figure 2**).

We tested the behavior of the female lethality construct at various concentrations of Tet (**Figure 2d–f**). When used in high concentrations, Tet has been reported to carryover between generations by maternal deposition in oocytes (**Schetelig et al., 2009**). In the presence of Tet at concentrations between 10 and 100 μg/ml there is a consistent but slight sex-ratio bias with 40–45% female offspring (**Figure 2d**). There is no statistically significant difference in the sex-ratio of offspring across these concentrations. When these offspring are transferred to food without Tet, the number of females that survive in the next generation varies from 0% to 35% in a dose-dependent manner ($R^2$ = 0.86 to best-fit Hill function, t-value = 24.3, p < 0.00001). Parents raised on 10 μg/ml had no surviving female offspring when transferred to Tet-free medium, but parents raised on 100 μg/ml had 35% female offspring on Tet-free medium (**Figure 2e**). We confirmed that this likely arises from maternal deposition of Tet and not genetic instability of the female lethal construct by rearing these flies for one additional generation in Tet-free medium. We observed 100% female lethality in the second generation of rearing on Tet-free media (**Figure 2f**).

## Male mating competitiveness and evidence for sperm displacement

We tested the ability of SSIMS males to compete with wild-type males for mating to wild-type females using two assays (**Figure 3a and b**). In the first, one male SSIMS fly, one male wild-type, and one virgin female were co-housed in a vial for 8 hr for mating. The female was then isolated to count

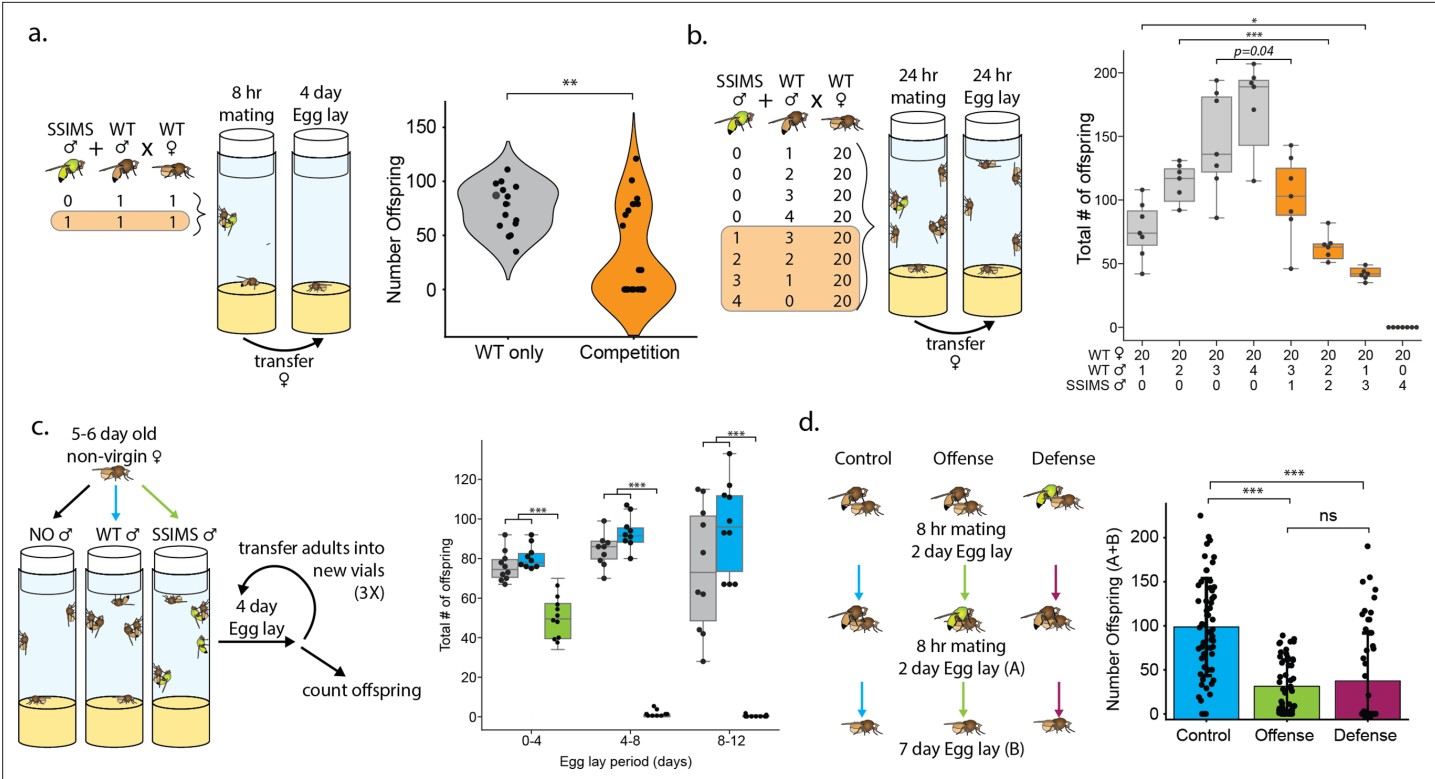

**Figure 3.** Male mating competition with wild-type *D. melanogaster*. (**a**) Experimental design schema and results of individual male mating competition assay. Violin plots show the number of offspring produced by one WT female crossed to one WT male (gray, n = 15) or one WT female crossed to one WT male and one SSIMS male (orange, n = 26). A Welch two-sample t test was performed between the two groups. (**b**) Experimental design schema and results of mixed-male mating assay. Box-whisker plots show the median (center line), 25th and 75th percentile boundaries (box), and min/max (whiskers). Student's t tests were performed between experiments with identical numbers of wild-type males. n = 7 per group. (**c**) Experimental design schema and results of sperm displacement assay. Box-whisker plots colored to show the addition of no males (gray), wild-type males (blue), or SSIMS males (green) to the non-virgin wild-type females. Student's t tests were performed for each egg-lay period comparing+ SSIMS male vs both no-male and WT-male groups. n = 10 per group. (**d**) Experimental design schema and results of offensive and defensive sperm displacement assay. Bar graphs show the number of offspring produced by one WT female mated to a WT male and remated with a second WT male (blue, n = 60), one WT female mated to a WT male and remated with a SSIMS male (green, n = 53), or one WT female mated to a SSIMS male and remated with a WT male (maroon, n = 54). Welch two-sample t tests were performed pairwise between each of the three groups. Statistical significance: ns = not significant, *=p < 0.01, **=p < 0.001, ***=p < 0.0001.

how many offspring she had in the next four days (*Figure 3a*). Females that had not mated during the 8 hr period (eggs laid by virgin females neither hatch nor necrose as do eggs laid by non-virgins) were not included in the final counts. In control experiments lacking the SSIMS male, the females had an average of 76 offspring surviving to adulthood. In the competitive experiments, there was a clear bimodal distribution. Nine females produced an average of 82 females for one mode, and fifteen females produced zero adult offspring. This latter group had clearly mated, because the females laid eggs that hatched into larva that died soon after. There were two females that produced an intermediate number of offspring, suggesting they may have mated with with both wild-type and SSIMS males. We performed a second competition experiment with an excess of females, similar to that reported previously (*Kandul et al., 2019*). Twenty virgin females were mated with four males at a 4:0, 3:1, 2:2, 1:3, or 0:4 ratio of wild-type to SSIMS. Control matings were also performed with twenty wild-type females and three, two, or one wild-type males and no SSIMS males to account for the decrease in offspring that is due to limited wild-type male availability when competing with SSIMS males. Fewer offspring survive when SSIMS males are included in the mating vials (*Figure 3b*), and this cannot be explained by the decrease in the number of males that could sire surviving offspring (p < 0.05 for each comparison).

In *Drosophila* spp., females can store the sperm from multiple males in their spermathecae which allows for sperm to compete to fertilize the eggs. We tested the hypothesis that sperm from SSIMS

males can displace wild-type sperm in the spermathecae of previously mated wild-type females. We placed three 5–6 day old non-virgin wild-type females (whose previous mating was with wild-type males) either with no males, three wild-type males, or three SSIMS males and measured total offspring of each group in 4-day windows for a total of 12 days (*Figure 3c*). Non-virgin wild-type females with either no males or with wild-type males for re-mating, produced an average of 70–90 offspring that survived to adulthood from each vial. While we did not empirically determine that females had mated during the initial 5–6 days, the assumption that females mated is supported by the negative control group (i.e. females moved to vials lacking males) which all produced offspring. When SSIMS males were available for re-mating, the number of surviving offspring dropped by 36% within the first egg lay period, and by 98–99% for the next two egg laying periods (*Figure 3c*). In the presence of SSIMS males, we observed post-zygotic lethality of eggs and larvae during all three egg-laying periods, which suggests that SSIMS males mated with the females to replace wild-type sperm with SSIMS sperm in the spermathecae. We performed a similar sperm displacement assay using shorter mating periods in both an offensive and defensive format, following methods described in previous studies (*Clark et al., 1995*). In both cases, the average number of surviving offspring following the second mating event decreased to one third the level of the control experiment (*Figure 3d*). There was not a

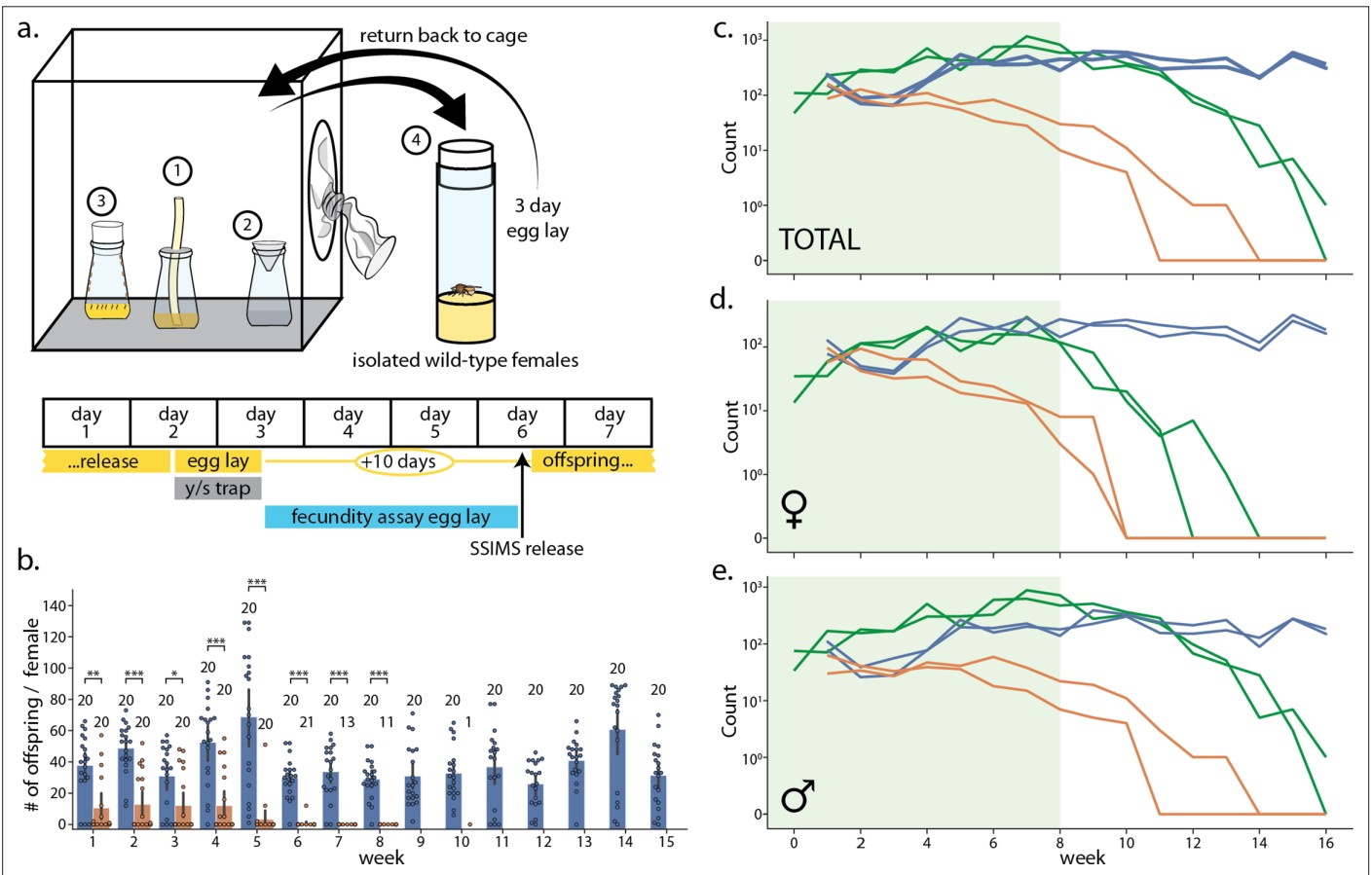

**Figure 4.** Laboratory cage trial of population suppression of wild-type *D. melanogaster*. (**a**) Cage trial design showing Bioquip cage with adult feeding apparatus (1), yeast/sugar trap (2), reproduction bottle (3), fecundity assay vials (4), and weekly schematic of the trial, with tasks done each day highlighted below. Two control and two SSIMS treated cages were initiated, see methods for details. (**b**) Results of the fecundity assay from each week. Each colored dot represents total number of offspring from a single vial with one wild-type female from control cages (blue) and SSIMS treated cages (orange). Sample size of each group per week is illustrated above groups. Statistical tests of significance (Mann-Whitney test) are performed for each week comparing control and +SSIMS treated cages, n = 11–20 per group. (**c–e**) Total (**c**), female only (**d**), and male only (**e**) counts from the yeast/sugar trap. Light green shading indicates weekly additional SSIMS releases until week 8. Wild-type population in the control cages (blue lines) rise gradually and reach plateau by week 6. Wild-type population in the SSIMS-treated cages (orange line) decreases gradually until wild-type females became undetectable by week 11. SSIMS populations in the treatment cages (green line) rise during weekly release periods, but fall rapidly after week 8 when releases cease. Statistical significance: *=p < 0.01, **=p < 0.001, ***=p < 0.0001.

significant difference in the number of offspring in offensive or defensive matings. This demonstrates that wild-type *D. melanogaster* will re-mate with SSIMS males in laboratory settings, and that the sperm from SSIMS males is competitive for fertilizing female eggs even in the presence of sperm from wild-type males in the spermathecae.

## Population suppression cage trials

We assessed whether SSIMS could be applied to suppress wild-type populations by conducting a cage trial with multi-generational populations of wild-type *D. melanogaster* with and without additions of SSIMS animals (*Figure 4a* and Supplementary Note 4). In cages, adults were sustained by feeding a liquid solution of 5% yeast extract and 5% sucrose solution via a cotton wick. This method of feeding adults was used to overcome the problem of adults getting stuck in semi-solid food containers. To allow the population to expand, a 250-ml bottle (reproduction bottle) with 30-ml standard cornmeal food was introduced to each cage once a week for 24 hr. After 24 hr, adults were removed from the bottle and released back into the cage. The bottles were capped to prevent additional egg-laying, larvae overcrowding, and adults getting stuck in the semi-solid medium. Reproduction bottles were uncapped 10 days after egg-laying and left open for 3 days to allow for the next generation to eclose and integrate with the multi-generational caged population. To assess population size and genotype, we placed an active-yeast/sucrose trap inside each cage once per week for 24 hr. We did not measure whether the fraction of total flies caught in the yeast/sucrose trap was density dependent, but the trapped counts did seem to correlate with total population size in the cage upon visual inspection. We also measured the fecundity of 10 randomly captured wild-type females from each cage once per week.

At the start of the experiment and for each of the first 8 weeks, we released 600 SSIMS flies to the two experimental cages (approximately 45:55 female:male ratio). The decision to stop at week 8 was arbitrary in this experiment. The released SSIMS flies had been reared on 100 μg/ml Tet, so we expected the female lethality phenotype from conspecific matings to be delayed by one generation (*Figure 2d–f*). On week 1, all four cages were seeded with 300 wild-type virgin females and 300 wild-type males. Weekly trapping, egg-laying, and fecundity assays were performed until week 16.

The SSIMS-treatment cages saw an immediate reduction of fecundity in wild-type females. Female fecundity did not follow a normal distribution, but rather was multimodal with a mode at 0 and another mode at about 50, likely representing females that had mated with SSIMS or wild-type males, respectively (*Figure 4b*). Several females gave low numbers of offspring, possibly due to re-mating with both wild-type and SSIMS males. By week 7, none of the captured wild-type females were fecund. The number of female fecundity assays that we could perform each week dropped as wild-type females could no longer be isolated from treatment cages.

*Figure 4c–e* show the results of the weekly trap counts. In the control cages (blue lines), counts of wild-type flies reached a steady-state of approximately 500 flies captured per week (*Figure 4c*). While cages were too large to allow for counting of all of the individuals, we estimate by visual inspection that the total cage population was approximately three times larger than the number of trapped/counted flies.

In the SSIMS treatment cages, the counts of wild-type flies dropped continuously throughout the experiment. While wild-type flies remained in each treatment cage by the time the SSIMS releases stopped in week eight, the high numbers of SSIMS males in the cages at that point prevented the wild-type population from recovering. By week 10, there were no wild-type females identified in the weekly traps.

The number of SSIMS flies, both males and females, remained high in the trap counts during weeks 1 to 8 when weekly releases occurred. This was expected as weekly releases of SSIMS adults occurred for the first 8 weeks. After SSIMS releases ended, counts of male and female SSIMS flies decreased, with female SSIMS flies decreasing rapidly to the point of disappearing as early as week 12. Male SSIMS flies disappeared from the cages a few weeks later. This is expected due to the FL construct allowing for one last generation of male-only offspring. By week 16, only a single male SSIMS fly was trapped across the two cages.

## Agent-based model of SSIMS releases

To investigate how SSIMS compares to other genetic biocontrol technologies, we developed an agent-based model for Spotted Wing *Drosophila* (SWD), an agricultural pest that is the target of genetic biocontrol research and development projects (*Asplen et al., 2015*; *Schetelig et al., 2021*). Details of the model are described in Supplementary Note 6.

We simulated genetic biocontrol by RIDL, FL, and three different rearing/release scenarios for SSIMS. The first scenario is a male-only release, which could be achieved by mass rearing SSIMS agents on 10 µg/ml Tet and hatching the released agents on Tet-free food. The second scenario is a single-amplification release, which could be achieved by mass rearing SSIMS agents on 10 µg/ml Tet and hatching the released agents on 10 µg/ml Tet food. This would be a bi-sex release in which one additional generation of male-only SSIMS biocontrol agents could be produced in the field if the released SSIMS males and females mate with each other. The third scenario is a double-amplification release, which could be achieved by mass rearing SSIMS agents on 100 µg/ml Tet and hatching the released agents on 100 µg/ml Tet food. This would be a bi-sex release that could mate to produce another bi-sex generation, but the subsequent generation would be male-only (*Figure 2d–f*).

We simulated a single growing season (March 1 - September 1) using historical temperature data for St. Paul, MN (*Figure 5*). Untreated populations expand to tens of thousands by early July before diminishing due to unfavorable temperatures. Each biocontrol agent was released across a combinatorial regime of release numbers and release frequencies and the total WT SWD that were hatched across the duration of the season were compared (*Figure 5b–e*). We also report days to eradication for each release strategy in *Figure 5f*.

There are several notable results from this modeling. First, FL and SSIMS generally outperform RIDL by providing more population suppression for the same number of release agents. The closest comparison to RIDL is the male only release of SSIMS. Because sex separation is automatic for SSIMS, we simulated 100% of the released biocontrol agents as male for this technique and 50% for RIDL. If manual sex separation were used to release only male RIDL agents, the behavior should mirror male-only SSIMS releases.

FL outperformed double-amplification SSIMS (100 mg/ml Tet) and single-amplification SSIMS (10 mg/ml Tet) at the lowest release numbers. This is due to the persistence of the FL genotype in the simulated population, even many generations after release. However, at moderate release numbers SSIMS outperformed FL. The largest advantage of SSIMS over FL comes when analyzing the time to eradication (*Figure 5f*), where SSIMS achieves population suppression faster than FL.

The simulations in *Figure 5b–f* include regular release of biocontrol agents throughout the entire growing season. A more realistic release scenario is an aggressive, but time-limited release early in the season. We simulated a weekly release of 80 biocontrol agents for four successive weeks during the month of April (*Figure 5g*). Bar graphs comparing the numbers of genetically modified and wild-type agents at June one and July one are shown in *Figure 5h and i*, respectively. For each release we plot the number of genetically modified and wild-type SWD throughout the growing season. Both RIDL and male only SSIMS persist only a short period after the releases end, as expected. In these simulations, wild-type SWD never reach as high of population levels as the non-treatment controls due to the lag time in population expansion caused by the early season suppression.

FL, single-amplification SSIMS, and double-amplification SSIMS all persisted longer than RIDL and male-only SSIMS and acheived better population suppression. Of these, FL persisted the longest and was not able to suppress the wild-type population completetly before the mid-summer peak. Both single- and double-amplification SSIMS eradicated the wild-type population by the end of June. The double-amplification SSIMS release agents persisted for an additional two weeks after the wild-type were eradicated. For single-amplificaiton SSIMS, both the GM and wild-type populations were eradicated by the end of June. It is worth noting that for both SSIMS strategies the last generation of GM agents is exclusively male, which do not damage fruit. While we did not simulate migration, the additional persistence of SSIMS males could potentially help combat small introductions later in the season.

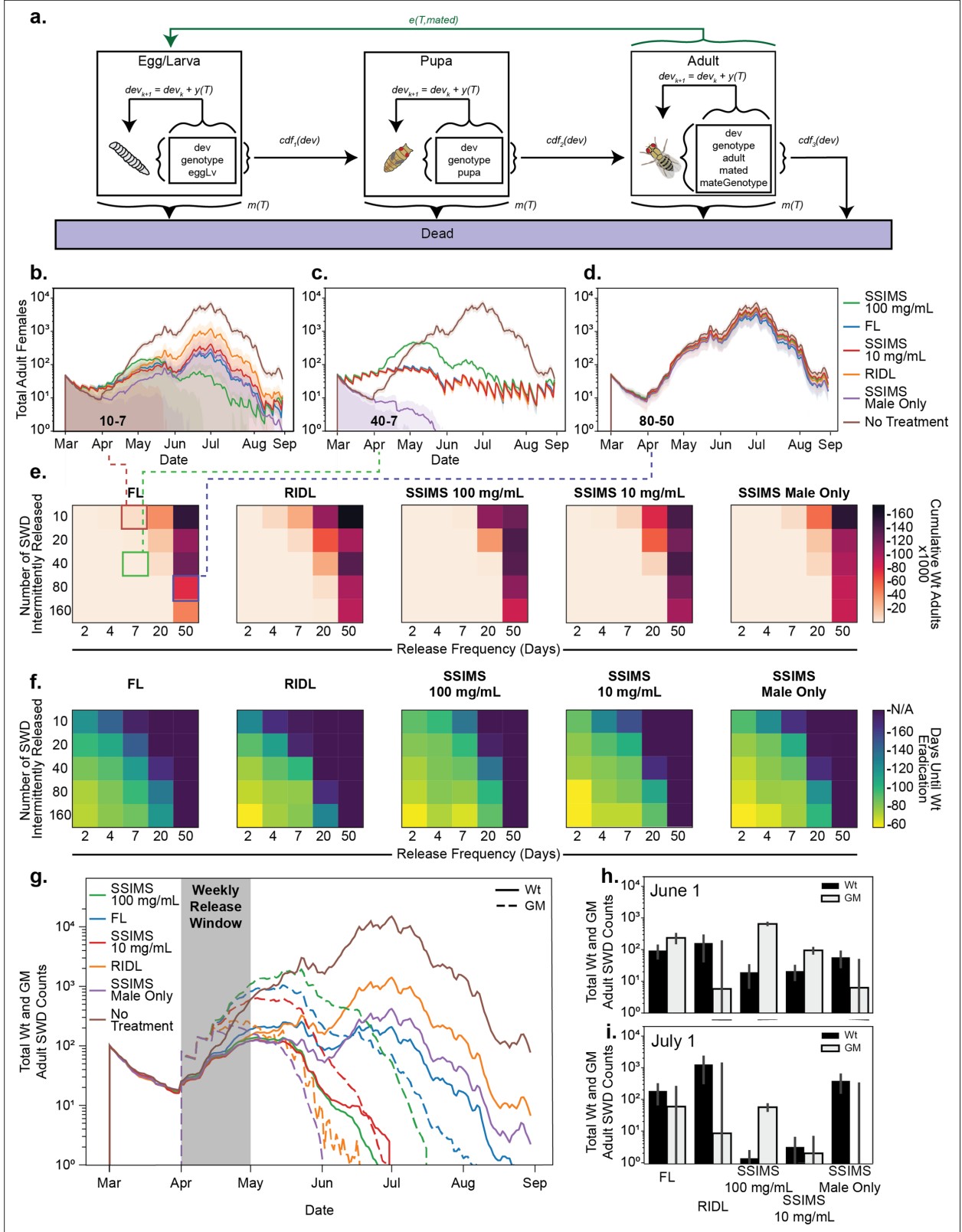

**Figure 5.** Agent-based modeling. (**a**) Schematic of model progression for an individual SWD agent. (**b–d**) Traces of total adult female populations over a growing season with different biocontrol agents. Shaded area represents one standard deviation among 10 simulations. Numbers inset to bottom left of plots show the insect release numbers and frequency (10–7 for (**b**), 40–7 for (**c**), and 80–50 for (**d**)) and correspond to grid cells identified with colored boxes in (**e**). (**e**) Heatmaps representing the cumulative total of WT adults over the course of the season at different release strategies. (**f**) Heatmaps

*Figure 5 continued on next page*

representing the the average time to Wt eradication. (**g**) GM agent release was limited to the month of April. Traces of total WT and GM populations were tracked over the season from 10 simulations. (**h–i**) Total counts of WT and GM populations at June one and July one time points for the April limited release. Error bars show standard deviation of the mean from 10 replicates.

## Discussion

Several new techniques for genetic biocontrol have been shown effective in proof-of-concept experiments (*Alphey and Bonsall, 2014*; *Maselko et al., 2020*; *Buchman et al., 2021*; *Buchman et al., 2018*; *Akbari et al., 2014*; *Champer et al., 2020*; *Gantz and Bier, 2015*; *Terradas et al., 2021*; *Windbichler et al., 2011*). Combination approaches that bring together two or more distinct techniques can exhibit synergistic behaviors that are attractive for genetic biocontrol applications. Here, we combine conditional female lethality with EGI to create the SSIMS strain.

The final genotype of the SSIMS flies generated in this study contains 23 synthetic genetic elements (operators, promoters, CDSs, terminators, etc.) spread across four chromosomal loci in the haploid genome. In total, 42,654 bases of synthetic DNA are integrated to the genome of engineered flies. This makes SSIMS one of the most complex engineered systems demonstrated in insects. This is exciting because it parallels the improvements in the complexity of engineered systems in other organisms and suggests that even more complex engineering will be possible with continued improvements of engineering methods. At the same time, the genetic complexity of SSIMS is a hurdle for its translation to pest organisms. Without the tools that we exploited in engineering SSIMS, such as multiple genetic markers and balancer chromosomes, engineering the SSIMS genotype in non-model insects will require more tedious molecular characterization of offspring produced during strain construction. However, we are encouraged by recent publications highlighting new genetic tools for non-model organisms (*Kandul et al., 2021*). Also, the molecular components needed to engineer SSIMS have been shown to work in diverse organisms.

Stacking more genetic components into a single organism can lead to fitness defects due to problems of resource allocation or cross-talk (*Cardinale and Arkin, 2012*; *Segall-Shapiro et al., 2014*). We observe a sex-ratio bias in favor of males when the SSIMS flies are reared with Tet, suggesting that the two X-linked female lethal constructs confer a slight fitness penalty in females even in the repressed state, presumably due to leaky or basal expression of tTa. This same bias was seen previously in the absence of the EGI constructs (*Das et al., 2020*). While slight sex-ratio biases like this might slow the expansion of a colony during mass rearing efforts, this is likely to be offset by the cost savings attained by automatic sex-separation or by the field amplification following bi-sex release. Based on previous observations with dosage effect of the female lethal construct, we predict that fine-tuning the expression strength of the tTA-regulated promoter will provide control over the dynamic range of the on/off state change. Here we favored strong penetrance at the expense of genetic burden, but this should be examined more carefully in species appropriate for field applications.

The competitiveness of biocontrol males with wild-type males is a critical factor in the efficacy and economy of SIT-like approaches. We show that the SSIMS males are competitive with wild-type males in laboratory mating experiments (*Figure 3a*), and that this extends to the level of sperm competition. This latter point is important for polyandrous insects. For example, *Drosophila suzukii* (Spotted Wing *Drosophila*, SWD) is closely related to *D. melanogaster* and is a global pest of fruit production. Several groups have observed female SWD re-mating in observational studies (*Revadi et al., 2015*; *Krüger et al., 2019*; *Clark et al., 2020*; *Lanouette et al., 2020*), and we have observed high levels of remating using a genetic assay (Supplementary Note 3). If a wild female mates with both a wild male and a SSIMS male, she will have a decrease in fecundity due to sperm displacement/competition. This would not occur for SIT-mimicking approaches that prevent sperm development in males. The laboratory competition assays performed here are promising but do not necessarily predict competition in the field. Suboptimal rearing conditions (*Dyck et al., 2005*), microbiome differences between wild and reared flies (*Sharon et al., 2010*), or behavioral differences like assortative mating (*Reisen et al., 1982*) could impact male mating competition. These factors should all be considered during the translation of an approach like SSIMS to field settings.

In its simplest application, SSIMS could be used to facilitate the sorting of male from female biocontrol agents prior to environmental release. In a male-only release, no biocontrol agents would persist beyond the released generation. Such an approach would be highly self-limiting, but would require

larger release numbers to achieve suppression. Alternatively, males and females could be released together to provide limited field amplification. In a bi-sex release, if either sex of SSIMS insect mates with wild-type, no offspring will survive. If two SSIMS insects mate with each other, more biocontrol agents will be produced to potentiate the population suppression.

Interestingly, we can control the persistence of the SSIMS genotype to a degree by tuning the amount of Tet present in the food during the stock maintenance phase (*Figure 2c–e*). When the SSIMS line is reared from eggs to adults in 10 µg/ml Tet and then the adults are transferred to food without Tet, 100% female lethality occurs in the F1 generation. However, higher concentrations of Tet (100 µg/ml) in the stock maintenance phase delays the female lethality by a generation, (i.e. we observe 100% female lethality in the F2 generation). This feature of the SSIMS line could be useful for field applications, where simply varying the amount of Tet during the stock maintenance phase and/or perhaps spraying Tet in the field could oscillate the state of SSIMS between persistence (high Tet, both males and females emerge to sustain SSIMS population) and self-limiting (no/low Tet, females die after two generations and SSIMS population collapses). Tet is already approved for use as a foliar spray during fruit cultivation.

Another benefit to the SSIMS approach is redundancy. Both EGI and conditional female lethality are biocontrol techniques in their own respect. If one component were to mutate to become inactive in a released SSIMS strain, the other component would still prevent long-term persistence in most scenarios. However, this redundancy is partially reduced by the fact that the components are not linked on the chromosome. Also, high levels of genetic resistance to tTA female lethality has been described in wild *Drosophila* populations (*Knudsen et al., 2020*).

There are several shortcomings of this study that will need to be addressed to determine the suitability of the SSIMS approach to real-world applications. First, our cage trial was not large enough to reveal low frequency resistance mutations that would undermine the population suppression effect. Such resistance mutations could (i) inactivate or silence expression of the PTA, (ii) introduce SNPs at the target site to prevent PTA binding, (iii) suppress the negative impact of ectopically expressing *pyramus* or an equivalent developmental morphogen, or (iv) produce assortative mating phenotypes that prevent wild-type females from mating with released males. We discuss and demonstrate methods to mitigate some of these resistance mechanisms in previous publications (*Maselko et al., 2017*; *Maselko et al., 2020*). Recently, published work describes instances where such resistance mutations can arise reproducibly at very low frequencies (*Zhao et al., 2020*).

This study also lacks a techno-economic analysis or feasibility study to determine whether insects can be mass-reared to sufficient numbers to allow applications of SSIMS. Such studies will be more appropriate when SSIMS has been demonstrated in a pest organism that has potential applications in biocontrol.

In conclusion, we present a proof of concept of SSIMS in a model insect, *D. melanogaster*. SSIMS represents a new genetic biocontrol approach that provides a unique set of strengths and weaknesses compared to other demonstrated or proposed strategies (*Alphey and Bonsall, 2014*). In species where release of genetically engineered females is acceptable, SSIMS affords a tunable field-amplification, while still displaying low persistence. The male biocontrol agents are competitive with wild-type males for copulation and their sperm is competitive within the spermatheca. This makes SSIMS an especially attractive self-limiting genetic control approach for polyandrous pest organisms.

## Materials and methods
### Fly stocks and rearing conditions

*D. melanogaster* strains were maintained at 25 °C and 12 hr light/dark cycle and 60–70% relative humidity. All experiments were conducted in standard cornmeal/agar food Bloomington Formula (Genesee Scientific, 66–121), 0.05 M propionic acid (Sigma, 402907), 0.1% Tegosept (Genesee Scientific, 20–258). Tetracycline was added to 70 °C cool fly food (100 µg/ml, unless specified) before pouring into vials. *White1118* strain was used as a wild-type. All crosses were performed with 2- to 5-day-old virgin females and males, unless specified.

## Generation of SSIMS line

SSIMS is generated by combining the sex-sorting female-lethal strain with the Engineered Genetic Incompatibility strains published previously (*Maselko et al., 2020*; *Das et al., 2020*). We made a compound stock that contains two copies of the female-lethal construct on the X-chromosome, a refactored *pyramus* promoter on the 2nd-chromosome, and a programmable transcriptional activator targeting the wild-type *pyramus* promoter (dCas9-VPR)PTA+(pyr)gRNA expression cassette on the 3rd-chromosome. The combination of the refactored promoter and PTA + gRNA garners the EGI capacity of the SSIMS line. The cross strategy used to combine the two transgenes is illustrated in *Appendix 1—figure 1*.

## Offspring lethality assays

Efficacy of female-lethality and EGI was measured by crossing 2- to 5-day-old SSIMS male and virgin females (SSIMS and wild-type) with or without Tet (10 µg/ml). Five virgin females and males were were allowed to mate in vials for 3 days, then removed to prevent overcrowding of larvae. Male and female offspring were counted 10 days post egg lay.

## Male competition assays

For the 1:1:1 competition assays (*Figure 3a*), 2- to 5-day-old flies (one wild-type male, one SSIMS male, and one virgin wild-type female) were placed in a vial with fresh media for 8 hr to mate. Females were removed to a new vial for a 4-day egg lay period. Females were confirmed to have mated by looking for eggs that hatched to larva. The number of offspring that survived to adulthood was counted 14 days post egg lay. To assess SSIMS male competitiveness various combinations of 0–4 SSIMS and wild-type males were placed together with 20 wild-type virgin females, and allowed to mate for 24 hr. After the mating period, males were discarded and females were transferred into a new vial for 24 hr egg lay period. Total offspring were counted 11 days post egg lay.

## Sperm displacement assay

To assess the ability of SSIMS sperm to displace wild-type sperm in utero (*Figure 3c*), three (5–6 day old) non-virgin wild-type females, which were previously mated with wild-type males, were placed in a vial either alone, with three wild-type male, or three SSIMS male. These males were left in the vials to remate with females for the remainder of the experiment. Females (with or without males) were allowed to lay eggs in 4-day periods then flipped into new vials for a total of 12 days. Total offspring were counted 11 days post egg lay for each period.

For the offense/defense sperm displacement assay (*Figure 3d*), virgin wild-type females were placed in a mating vial with males (wild-type or SSIMS) for 8 hr and then moved to individual tubes for 2 days. Females that showed evidence of mating (eggs hatching to larva) were then mated with the other type of male for 8 hr. Females were moved to a fresh vial to lay eggs for 2 days and then transferred to a final vial for an additional 7 days of egg laying. We report the number of surviving adult flies from the final 2 day and 7 day egg lay vials. The number of offspring that survived to adulthood was counted 14 days post egg lay.

To investigate polyandry and sperm competition in *D. suzukii*, two inbred SWD lines, with unique fixed SNPs in the 5' promoter region of the *wg* gene, were crossed. Mating pairs of 3 non-virgin 'SWD-A' females with 0 or 2 males of 'SWD-A' or 'SWD-B' were placed in vials. Genomic DNA of adult offspring were isolated as previously described (*Das et al., 2020*). PCR of the *wg* promoter region was done using primers wgF (5'-AGATTGCGCAAATAATCCGGC-3') and wgR (5'-ATTCGAGC-GGAGGAGTGAAG-3'), and Q5 polymerase following manufacturer recommendations(New England Biolabs). PCR products were Sanger sequenced by ACGT (Wheeling, IL), and the results were analyzed by Synthego's ICE software (ice.synthego.com).

## Cage trial

Cage trial experiments were designed to simulate multi-generational and generational-overlapping populations dynamics of wild-type *Drosophila*. On week 1, the cage trial was initiated with 600 (50% female) wild-type virgin males and females into each cage. A total of 600 SSIMS adults (40–50% female) were released weekly in the+ SSIMS treatment cage. Adult populations were housed in 0.027 m³ cages (Bioquip, 1452) and were fed by supplying 50 ml of 5% yeast-extract, 5% sucrose (YES), 0.1%

Tegosept solution via a cotton wick in a 250 ml standard fly bottle and wrapped at the rim to prevent flies from getting stuck in the bottle. YES solution was replaced weekly. To allow the population to grow, 250 ml bottle (reproduction bottle) with 30 ml standard CMF were introduced once a week for a 24 hr egg lay period, after which the adults were removed. Reproduction bottle were uncapped 10 days after egg-lay and left open for 3 days to allow for the next generation to eclose and mix/ mate with the preexisting population in the cage. To assess population size and genotype, an adult fly trap with yeast/sugar (YS) solution was placed in the cage once a week for 24 hr. The fly trap was baited with 30 ml of YS solution consisting of 0.5% active-yeast, 2.5% sucrose, 0.5% Micro90 (Cole Parmer). To measure fecundity of wild-type females, once a week, random adults from each cage were trapped in CMF bottle and wild-type females were isolated individually into 10 separate vials, and the remaining adults were released back in the cage. Wild-type females were allowed to lay eggs in vials for 3 days then released back in the cage. Total offsprings from each vial was quantified 11 days post egg-lay.

## Data analysis, statistics, and visualization

All experiments were performed at least twice except the cage trial. Each batch of experiments contained at least five replicates, see caption for each Figure. Chi-squared test was performed to test difference between observed and expected sex ratio for *Figure 2b* and student's t-test for *Figure 2c*. For *Figure 3a/d*, a Welch's t-test was applied. For *Figure 3b and a* one-way ANOVA with Tukey post hoc test for multiple comparisons was applied. For *Figure 3c and a* Student's t-test was applied. For *Figure 4b*, Mann-Whitney non parametric test was applied. Data was analyzed in Python using pandas, scipy, and statsmodels libraries. Statistical significance: $*=p < 0.01$, $**=p < 0.001$, $***=p < 0.0001$. Graphs were generated using the Seaborn and Matplotlib libraries. Graphs were modified in Illustrator to fit publisher requirements.

## Acknowledgements

We thank Dr. Max Scott for helpful discussions about conditional female lethality and for providing us with genetic reagents. We thank Dr. Michael O'Connor for balancer strains that were used for the cross strategy. MJS, AU, SD, and MM were supported in part by the Defense Advanced Research Projects Agency (grant number D17AP00028). The views, opinions, and/or findings contained in this article are those of the authors and should not be interpreted as representing the official views or policies, either expressed or implied, of the Defense Advanced Research Projects Agency or the Department of Defense. AU and NF were supported by funding provided by the Minnesota Invasive Terrestrial Plants and Pests Center through the Minnesota Environment and Natural Resources Trust Fund.

## Additional information

### Competing interests

Ambuj Upadhyay: Inventor of filed patents (PCT/US2019/059826). Nathan R Feltman: Inventor of filed patents.(PCT/US2019/059826). Siba R Das: Inventor on filed IP, co-founder of Novoclade. (PCT/US2019/059826). Maciej Maselko: Inventor of filed IP; co-founder of Novoclade. (PCT/US2019/059826). Michael Smanski: Inventor on filed patents and co-founder of Novoclade. (PCT/US2019/059826). The other authors declare that no competing interests exist.

### Funding

| Funder | Grant reference number | Author |
|---|---|---|
| University of Minnesota | | Michael Smanski |
| Defense Advanced Research Projects Agency | | Michael Smanski |

| Funder | Grant reference number | Author |
| --- | --- | --- |
| Minnesota Invasive Terrestrial Plants and Pests Center | | Michael Smanski |

The funders had no role in study design, data collection and interpretation, or the decision to submit the work for publication.

## Author contributions

Ambuj Upadhyay, Conceptualization, Formal analysis, Investigation, Methodology, Visualization, Writing – original draft; Nathan R Feltman, Formal analysis, Investigation, Methodology, Validation, Visualization, Writing – review and editing; Adam Sychla, Investigation, Writing – review and editing; Anna Janzen, Formal analysis, Investigation, Writing – review and editing; Siba R Das, Maciej Maselko, Conceptualization, Methodology, Writing – review and editing; Michael Smanski, Conceptualization, Formal analysis, Funding acquisition, Project administration, Supervision, Visualization, Writing – original draft, Writing – review and editing

## Author ORCIDs

Michael Smanski http://orcid.org/0000-0002-6029-8326

## Ethics

Work with invertebrates (e.g. D. melanogaster) is exempt from the University of Minnesota's IACUC research oversight, however all work was approved by UMN's Institutional Biosafety Committee.

## Decision letter and Author response

Decision letter https://doi.org/10.7554/eLife.71230.sa1
Author response https://doi.org/10.7554/eLife.71230.sa2

---

## Additional files

### Supplementary files

• Transparent reporting form

### Data availability

All data is available in the manuscript.

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

## Appendix 1

### Supplementary Note 1

**[see Appendix 1 Figure 1]**

### Supplementary Note 2

**[see Appendix 1 Figure 2]**

### Supplementary Note 3

[See *Appendix 1—figure 3*]. We tested whether *Drosophila suzukii* (Spotted Wing *Drosophila*), an agricultural pest, was similarly polyandrous in a laboratory mating assay. The experiment was similar to the sperm competition assay described in *Figure 3c*, but males from two different inbred populations were used. Strain 'SWD-A' females were placed in vials with either no males, 'SWD-A' males, or 'SWD-B' males. The SWD-B population has unique fixed single nucleotide polymorphisms (SNPs) that allow us to track paternal identity of the offspring easily using Sanger sequence analysis (*Appendix 1—figure 3*). From the first egg-lay period, 3 replicates had a SNP signature that suggests re-mating. From the second and third egg-lay period, four out of five and three out of five replicates respectively had re-mated (*Appendix 1—figure 3*). These results support previously reported observational studies of re-mating in SWD (*Revadi et al., 2015*; *Krüger et al., 2019*; *Lanouette et al., 2020*; *Clark et al., 2020*). Our observed re-mating rates are higher than those reported, because our assay design allowed of continuous co-housing of previously mated females with males. Previous studies only looked for evidence of re-mating in short observations windows each day. Our use of amplicon sequencing as a measure of re-mating frequency is the first of its kind to our knowledge and allows for re-mating events to be indirectly observed over a continuous period that last many days.

### Supplementary Note 4

**[See Appendix 1 Figure 4]**

### Supplementary Note 5

**[See Appendix 1 Figure 5]**

### Supplementary Note 6: Agent-based model of Spotted Wing *Drosophila*

The ODD protocol was developed as a field-specific standard for communicating agent-based models (*Grimm et al., 2006*). In addition to describing our model in this format, we have made the source code publicly available via GitHub (https://github.com/smanskiLab/SWD_Agent_Model) to allow for reproducibility of other groups who would like to make use of this model.

#### Purpose

We developed an agent-based model for *Drosophila suzukii* in Python (*Van Rossum and Al, 2010*) that allows us to monitor the performance of various genetic population control strategies. We built the model using the Mesa platform (*Kazil et al., 2020*). This agent-based model allows us to observe systems-level (population-level) properties that emerge from changes in the attributes of individuals. We chose an agent-based model instead of a mathematical modeling by ordinary differential equations (ODEs) as the former better captures complex behavior, particularly in population modeling where stochastic events in a small number of agents (*i.e.* evolution of resistance) factors heavily into the observable, population-level outcomes. Since we explore population suppression methods, most simulations at some point arrive at populations comprising very small numbers of individuals. Stochastic events in small populations can lead to population instabilities that are better captured with stochastic agent-based models than ODE-based models (*Lacy, 2000*). The model we deploy here is a simple agent-based model that considers three life-stages: (i) egg/larval, (ii) pupal, and (iii) adult. We have designed our model so that it can run with limited computing resources, for

example a typical run with 100 seeded flies that generates a population size of 70,000 can be run in minutes on a personal laptop computer. We incorporate historical temperature data from March 1st, 2010 to calculate temperature dependent mortality rates and developmental rates. We do not have a density dependent mortality, and we do not try to estimate food abundance. Both of these are likely to impact population levels, but there is not sufficient data available to be able to simulate that impact. We used empirical data for temperature dependent mortality, temperature-dependent development, and temperature-dependent fecundity. Flies in our model re-mate daily and maintain the sperm from their last three mating partners in their spermathecae such that each new egg has a chance of being fertilized with the sperm deposited by recent mating partners. The most important aspect of our model for the sake of this manuscript is our approach to capturing the behavior of genetic biocontrol agents. This and other details of the model are described in detail below.

## State variables and scales

Our agent based model was built in the Mesa platform with a single model class (named 'SWDModel') populated by a single agent type (named 'SWD'). The 'SWD' class contains attributes that are either fixed for the duration of a simulation or are updated with each time step. Invariant attributes of 'SWDModel' include the grid height and width (corresponding to a two-dimensional map of the relative location of neighboring populations), the type of scheduler ('RandomActivation', so that agents states are updated in a unique and random order during each timestep), the migration rate between grid cells, and the data collector, which by default is programmed to report data on the sex, life-stage, mating status, genotype, and position of each agent at every timestep. While these attributes are invariant over the course of a simulation, they can be adjusted any time the code is run. The variant attributes of 'SWDModel' that are updated with each timestep include the number of agents, the number of new agents, the temperature, and each of the temperature-dependent variables.

As alluded to in reference to the data collector, the 'SWD' agents contain a number of state attributes, including genotype, state (an alphanumeric representation of life-stage that facilitates identifying agents that are eggs/larva, pupae, or adults), sex, mating history, mate genotype, and position in the grid. A list is kept for each female agent recording the possible sperm genotypes from her last three mating partners (representing sperm present in the spermathecae). For egg fertilization, paternally-contributed genotypes are pulled from this list in a manner that favors the most recent mating partners (70%, 20%, 10% for past three mating partners).

## Process overview and scheduling

The model proceeds in time steps of 1 day. Next, the data collector runs and records the attributes for each agent in the model. New values for the number of agents at each lifestage are updated and temperature data is used to recalculate temperature-dependent variables including egg-laying rate and developmental rates. After updating the model-level attributes, individuals are randomly activated and processed in the following order. Adult females have the ability to remate daily, and sperm from the last three mating partners is stored in the spermathecae. Females use the stored sperm to lay eggs in a temperature-dependent manner. All agents have a numerical developmental timer that starts at 0 in each lifestage. Each timestep, the numerical value is increased based on the temperature dependent degree-day model, and agents progress to the next lifestage following a Gaussian curve with a mean and standard deviation unique to each lifestage (*Asplen et al., 2015*). All adults are given the opportunity to migrate to a neighboring population (stochastic). For this study we did not look at the impact of migration. A temperature-dependent mortality factor randomly removes agents of any lifestage. Any agents that die during a time step are removed from the scheduler. Once the scheduler has processed every agent in the model, the next time step proceeds and the process is repeated.

## Design concepts

### Emergence

The robustness of a particular genetic biocontrol technique to the counteracting pressures of genetic resistance is an emergent property in the model. While the resistance rate is imposed via parameter settings, the minimum resistant rate at which a particular biocontrol approach is successful at suppressing a population arises from interactions of other model properties including aggressiveness of release rates, migration rates between sub-populations, and the genetics of biocontrol agents.

## Fitness, fecundity and competitiveness

The model does not currently simulate different mating competitiveness, although could easily be added to the code. We model fecundity for each female indirectly using a temperature-dependent egg-laying and mortality rates. Females that live long enough will all have a lifetime fecundity of 200 eggs. We do not currently model unique fecundity or longevity variables for different genotypes of SWD as empirical data that would guide parameter set-points are not available for most technologies. The exception to this is fitness penalties that arise from genotypes of offspring from mating events. For example, inviability of hybrid offspring between wildtype and SSIMS flies impacts the fitness of both. These fitness costs are not directly specified anywhere in the model, but come out of the genetic rules governing embryonic lethality that are implemented during the egg/larval stage.

## Interactions

Our model assumes random mating between adult female SWD and any adult male SWD in the same grid cell (*i.e.* sub-population). Our model is written in a way so that it can be adjusted to account for things like assortative mating in the future, but that is not a current feature.

## Stochasticity

Assortment of genotypes between parents and offspring is governed by random chance, with likelihoods and genetic linkage set as user-defined parameters. Additionally, the survival of agents from one life-stage to the next is determined stochastically. Development times for each life stage follows a normal distribution around empirically measured averages (*Asplen et al., 2015*). The focus of this model is on population-level phenomena that emerge from these stochastic events.

## Observation

State variables are collected for each individual at every time step. As the size of the resulting data collector file can be substantial, we recommend 'commenting out' any state variables that will not be used in subsequent data analysis prior to running large simulations.

## Initialization

This model is meant to simulate SWD populations in environments similar to that seen at our field sites in Minnesota. SWD populations at these sites do not reach equilibrium, but are characterized by discrete spikes that coincide with optimal temperature ranges. The first flies start showing up in traps each year in early summer.

Each grid is seeded with a wild-type population of 100 adults. Since our model does not have density dependent factors, increasing the starting population is an approximately a linear transformation of the simulation (i.e. doubling the input, doubles the output). Because temperatures for population growth are not reached until April, there is an initial decrease in population. We do not recommend starting with less than 100 individuals, as the probability that the population stochastically will increase noise dramatically. Starting with more than 100 individuals is acceptable, but will increase computational resources needed to complete the run. For genetic biocontrol, GMO agents with the genotype that simulates RIDL, FL, or SSIMS are added to the population in numbers and at a frequency that are set prior to running the simulation. These are also released as adults. Temperature data is added to the model from historical 2010 daily high temperatures recorded in Minneapolis. We did not verify that the model accurately captures population behavior when run with temperature data from other locations, so users should be cautious when substituting temperature data.

## Input

One file is needs to be included as an input to initialize the simulation. It is a.csv file ("Temp_input. csv") containing temperature data for the simulation period. This file should be organized with two column headers in row 1, "Temp" and "Date". Daily high temperatures should be listed in Celsius. Dates are included in this input file for reference only and are not fed into the model. The code is designed to run from the operating system command line and takes three additional arguments as input: GMO release frequency, GMO release size, and an output directory. For example, running a simulation releasing 40 GM flies every 7 days, depositing output files into the subfolder "Run0" would be written in the form: "python 2021-10-05_SingleGenome_SWD_Cleaned_AS.py 7 40 Run0/".

## Output

Two files are output for each simulation. A.csv file is returned containing the cumulative counts of flies at each timestep. This file has data on the cumulative totals for GM, Wt, both, female GM, female Wt, and total females. A.pkl file is output containing state information (i.e. genotype, sex, lifestage) about every agent at every timestep. Running the "ArbGT_csvFileMakervSSIMS.py" publicly available via GitHub (https://github.com/smanskiLab/SWD_Agent_Model; copy archived at swh:1:rev:83aa66850ff14305f81556c2f8ca9a5ac2557748; *Smanski, 2022*), will read each.pkl file in the directory and return.csv files containing total number of flies alive at each timestep, with additional resolved by genotype and lifestage.

## Submodels

A list of the parameters used in the agent-based model are given in *Appendix 1—table 1*. Below is a more detailed description of the processes that take place during a simulation.

## Life-stage progression

Within a single population, individual agents in our model progress through a series of three discrete states (life-stages) in time-steps of 1 days. These states include egg-larval, pupal, and adult states. At each state, an individual's development is set to 0. During each time step in a simulation, the development progresses forward in a temperature dependent manner adopted from *Asplen et al., 2015*.

$$y_{k+1} = y_k + 234 \times \frac{0.0044 \times (T - 5.975)}{(1 + 4.5^{T-31})}$$

where y is the number of degree days and $t$ is the daily high temperature in degrees Celsius. Agents progress to the next state (e.g. pupa to adult) in a probabilistic manner if a randomly generated number between 0 and 1 is greater than the cumulative distribution function (cdf(y)) of the normal distribution:

$$cdf(y) = \frac{1}{2} \times (1 + erf(\frac{y - \Lambda}{\sigma\sqrt{2}}))$$

where y is the number of in a given life stage, A is the mean degree days at which an agent progresses to the next life stage and $\sigma$ is the standard deviation around this mean. These averages and standard deviations are unique to each lifestage and were previously reported (*Asplen et al., 2015*) A temperature dependent mortality rate taken from empirical data:

$$m(T) = 0.00035 \times (T - 15)^2 + 0.01$$

where $t$ is the daily high temperature is applied to all agents (*Asplen et al., 2015*).

Each female is limited to a maximum of 150 eggs (*Tochen et al., 2014*). In a given time-step, the number of eggs laid by mature females is determined by the temperature-dependent function below. We used the following polynomial function to fit previously reported experimental data (cite https://doi.org/10.1093/jee/tow006):

$$e(T) = \alpha \left[ \frac{\gamma + 1}{\pi\lambda^{2\gamma+2}}(\lambda^2 - ([T - \tau]^2 + \sigma^2))^\gamma \right]$$

where $\alpha = 659.06$, $\gamma = 88.53$, $\lambda = 52.32,$, $\sigma = 6.06$, $\tau = 22.87$, and $T$ is the temperature in degrees Celsius. The result of this function is rounded down or up to the nearest integer with a weighted stochastic function. For example an egg-laying rate of 1.5 eggs per day would be stochastically rounded up to two or down to 1 with a 1:1 odds ratio. An egg laying rate of 1.9 would be rounded up to two or down to 1 with a 9:1 odds ratio. [See *Appendix 1—table 1*].

## Mating

Adult females with a genotype that allows them to be fertile mate with a random male in the population during early adulthood. The males from these matings can include any living adult male. Adult females are allowed to mate each day, and sperm from the three previous mating partners is stored in a list and used to fertilize eggs with a 7:2:1 odds ratio, with the more recent mating partners' sperm preferred. During each fertilization event, eggs are generated by random assortment of alleles

provided by the maternal and paternal genotypes. Our model has a linkage disequilibrium table that allows us to link alleles computationally, for example to simulate separate genetic constructs that were intentionally placed close together on the same chromosome. [See *Appendix 1—table 2*].

## Genotypes

The genetics of FL, RIDL, SSIMS, and other biocontrol approaches not utilized in experiments for this study are encoded as a string of characters associated with each agent in the simulation. Definitions of each character are given in *Appendix 1—table 1*. For this study, we only use the *pptt* locus to simulate EGI. Female Lethality is simulated with the *ll* locus (*Ll* and *LL* would be heterozygous or homozygous for the FL allele, respectively). RIDL is simulated the same as FL but is set to be lethal in both sexes. SSIMS is simulated by combining the EGI and FL genotypes. The penetrance of the FL phenotype, which we can control by tuning the amount of tetricycline in the rearing media, is described in the model by the boolean 'carry' variable.

Some of the allele types reported in *Appendix 1—table 1* are included here for complete description of the model, but were not used in simulations described here. For example, the *bbdd* loci, which would allow for simulations of EGI with two independent target loci, are fixed as wild-type alleles for each agent in our simulations. Similarly, the alleles that would allow for modeling homing gene drives (W, G, R, I) are fixed as W for simulations reported here. Our model has the ability to model several mechanisms of resistance (described below), but those simulations were beyond the scope of this study. The starting genotype of wildtype or biocontrol agents released in simulations reported in this study are given in the table below. [See *Appendix 1—table 3*].

## Genetic resistance

We model promoter conversion events in EGI or SSIMS populations during the reproduction step. In any new egg that would be heterozygous at the promoter location (i.e. Pp or pP), the wild-type 'p' allele is changed to the resistant 'q' allele stochastically at a promoter conversion frequency specified at the start of the simulation. Because the rules for embryonic lethality require the co-existence of 'T' and 'p' alleles, these 'q' mutants remain viable. Functionally, this is equivalent to replacing the 'p' for an engineered 'P', but the 'q' designation facilitates future tracking of the mutated allele. An equivalent process occurs for the wild-type 'b' to resistant 'z' conversion. We model resistance in the SGD simulations at the stage of gametogenesis. For gene drive carriers ('WG' or 'GW'), the 'G' allele is passed on at a frequency of:

$$F_G = 0.5 + 0.5(F_{homing}(1 - F_{NHEJ}))$$

where $F^{homing}$ is the homing frequency and $F_{NHEJ}$ is the frequency of non-homologous end joining. Best case gene drive simulations use a homing frequency of 100% and a non-homologous end joining frequency of 0%. The gamete will inherit the G allele from a heterozygous parent stochastically if a random number between 0 and 1 is less than or equal to FG. Our model simulates non-homologous end joining if the random number generated, N, lies in the following range:

$$0.5 + 0.5(F_{homing}(1 - F_{NHEJ})) < 0.5 + 0.5(F_{homing})$$

In this case, a resistant allele 'R' is passed on one third of the time and a haploinsufficient allele 'I' is passed on two thirds of the time. These probabilities are based on the random likelihood that an indel generated by NHEJ will result in an in-frame coding DNA sequence (CDS). Our model assumes that any NHEJ repair that generates an in-frame CDS will still be haplosufficient and will resist further cutting by the drive nuclease. This is likely a slight overestimate of resistance given a particular NHEJ frequency, but is reasonable for the purposes of modeling.

## Timing of lethality arising from genetic control

For mating events that produce embryonic lethality, inviable embryos are removed from the model at the egg/larval lifestage. These mating events include those between EGI and wild-type agents, or more specifically, any egg whose genotype contains at least one allele 'p' and one allele 'T'. Likewise, eggs with no copies of the haplosufficient gene targeted by the gene drive (genotypes 'GG', 'GI', or 'IG') die at the egg/larval lifestage. While sex-ratio biasing approaches have been published that effectively remove females from the mating population at adult lifestages (e.g., Alphey's flightless female approach, cite), here we model embryonic lethality for both RIDL and female lethal approaches, as this matches the molecular biology in our system. In general, the timing

of lethality in this SWD model should not impact model results, as we do not simulate density-dependent mortality at early lifestages.

## Movement in multi-population models

For simulations with multiple neighboring populations, each adult agent is given the opportunity to move to an adjacent population at every time step. The migration rate (range [0–1]) is a user-defined parameter that determines the likelihood of movement between populations. Migrations can only occur to a cell that is adjacent or diagonal to the current position. We did not turn this feature on in the current study.

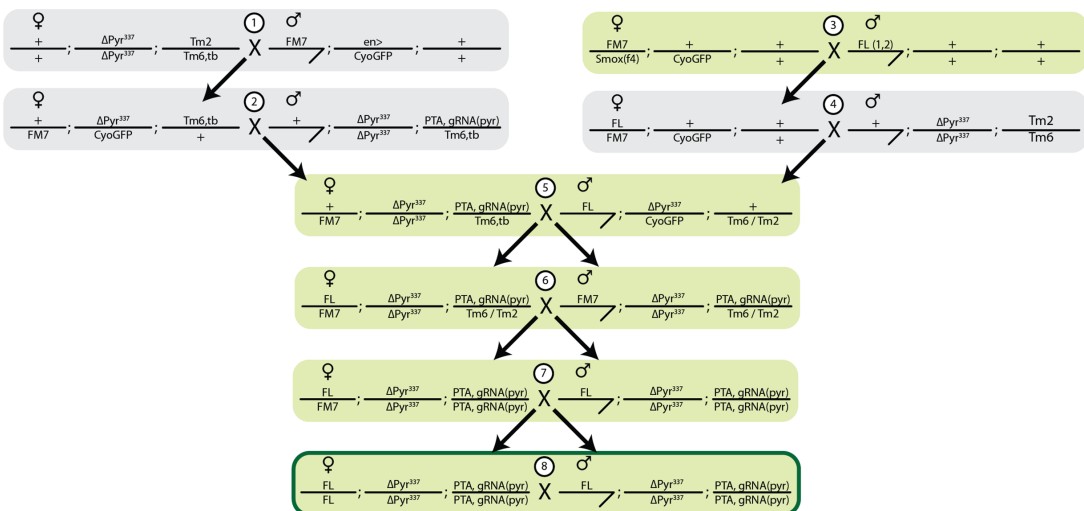

**Appendix 1—figure 1.** Cross strategy to generate SSIMS line. Female lethal (FL) and EGI (pyr337; PTA, gRNA-pyr) were crossed to balancer chromosomes, and subsequent males and females were isolated indicated by the black arrow. FL chromosomes contains 2 insertions on the X chromosomes, homozygous females contain 4 copies of the transgene. PTA+gRNA-pyr is a single insertion site containing foxo driven dead-Cas9::VPR and 2 guide RNAs driven by U6:3 and U6:1 promoters. Green shading indicates crosses done in the presence of 100 g/ml Tet. FL was tracked by GFP, PTA+gRNA was tracked by red eyes (mini white), and the pyr337 promoter mutation was tracked using balancers.

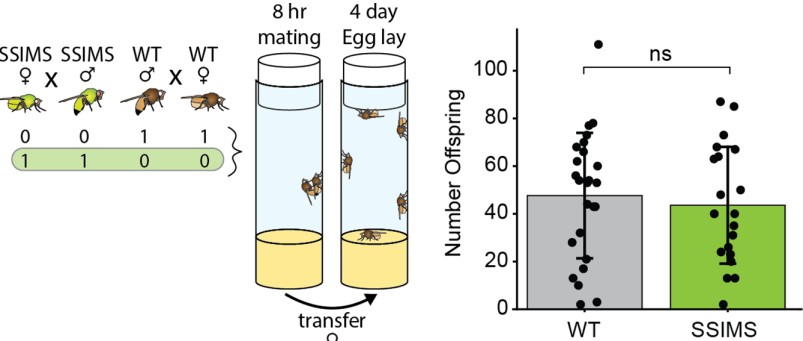

**Appendix 1—figure 2.** Fecundity of SSIMS flies. Experimental design schema and results of fecundity assay. Bar graphs show the number of offspring produced by one WT male crossed to one WT female (gray, n = 25) or with one SSIMS male crossed to one SSIMS female (green, n = 20). Welch two-sample t test was performed between the two groups. ns = not significant.

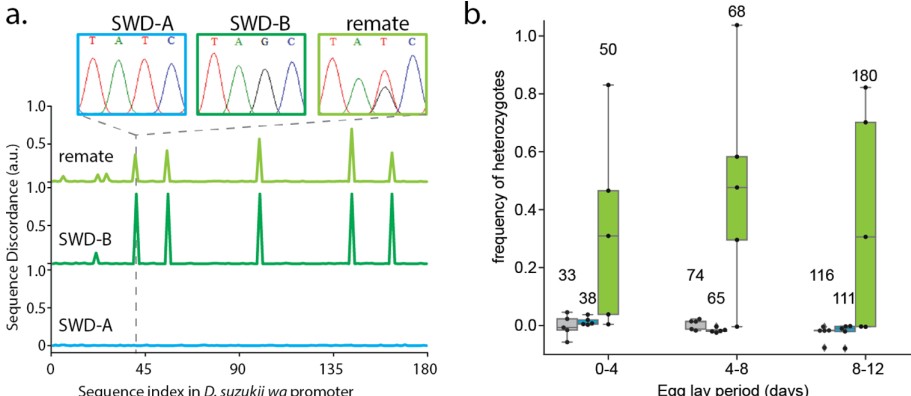

**Appendix 1—figure 3.** Polyandry in Spotted Wind Drosophila. (a) Representative data highlighting the calculation of sequence discordance when compared to the wg promoter sequence of inbred strain SWD-A (blue). Inset Sanger sequence chromatographs highlight the T41G SNP in inbred strain SWD-B (dark green). The remating trace (light green) is representative of many samples from (b). (b) Heterozygote frequency when non-virgin SWD-A flies were co-housed with no males (grey), SWD-A males (blue), or SWD-B males (light green). Numbers above box-whisker plots show total number of offspring collected from each experimental group. A one-tailed Welch's T-test of significance was performed for each egg-lay period comparing+ SWD B male vs+ SWD A male groups (n = 5 per group), and each produced a p-value between.01 and.05.

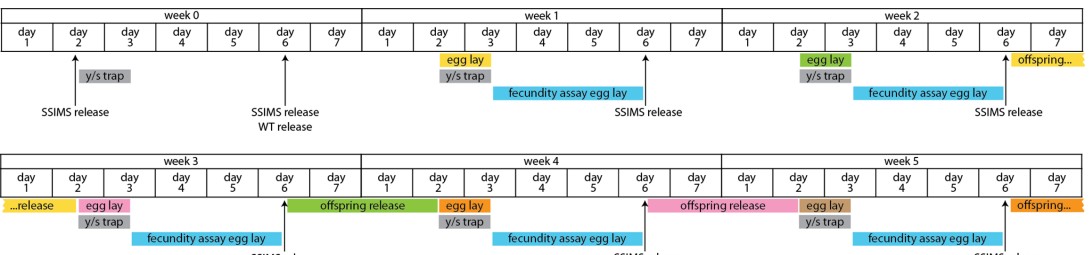

**Appendix 1—figure 4.** Extended timeline of cage trial design. In this multi-week view of the cage trial data, the timing between egg lay and offspring release is shown by coloring corresponding events. During the 10 days between egg lay and offspring release, the bottles were kept in cages, but were capped with a foam plug to prevent caged flies from entering and becoming stuck in the cornmeal food.

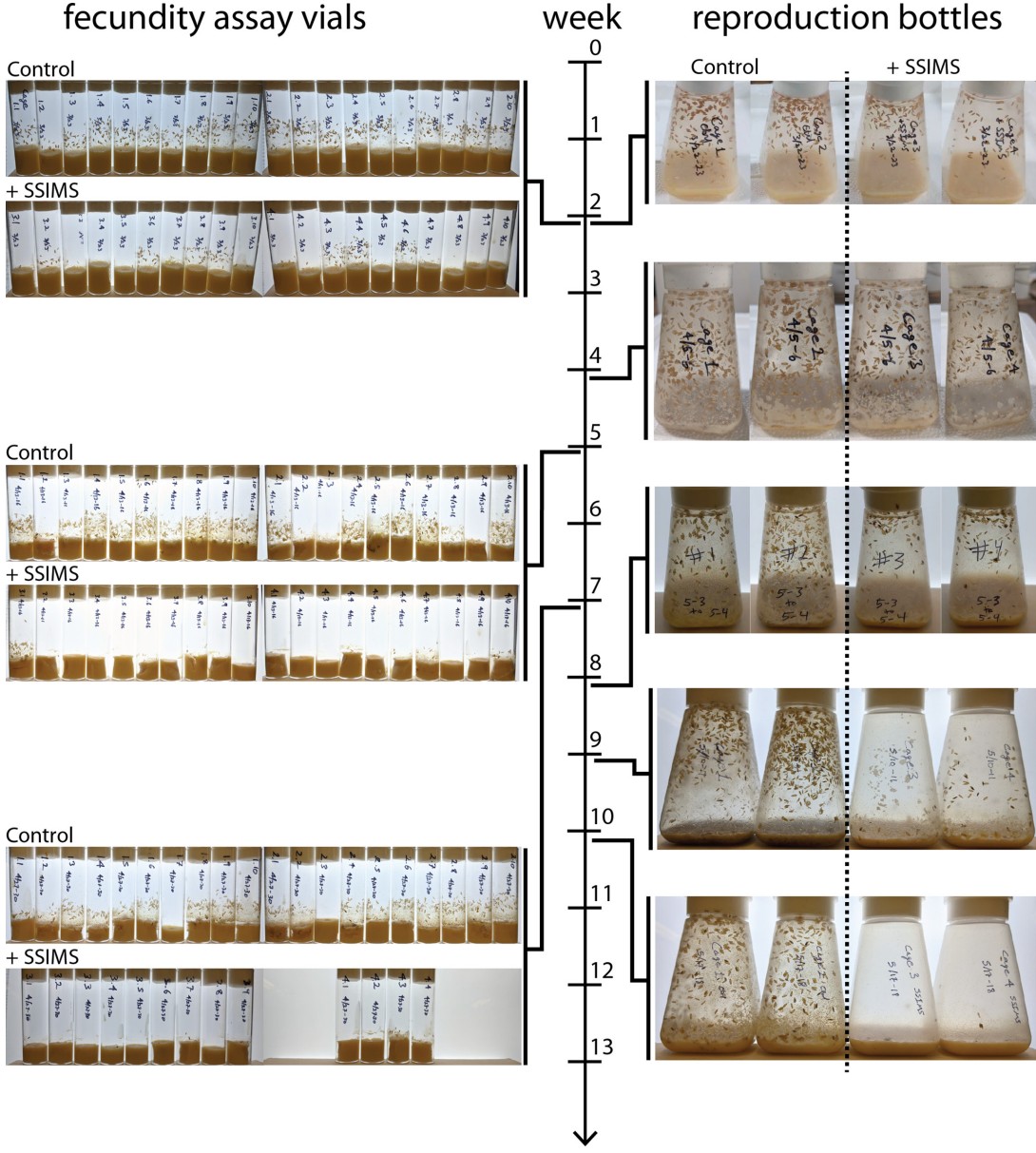

**Appendix 1—figure 5.** Sample pictures from cage trial. Timeline of cage trial shown in middle. Jagged lines indicate the week when images were taken. Fecundity assay vials from control cages and SSIMS treated cages to the left of the timeline. Images of the reproduction bottles from the control and SSIMS treated cages to the right of the timeline.

**Appendix 1—table 1.** Key parameters in SWD model.

| Parameter | Value | Reference |
| --- | --- | --- |
| Maximum Egg/Female | 150 | Tochen |
| Egg/Larval Degree Days - mean | 140.785 | Asplen |
| Egg/Larval Degree Days - SD | 20.49 | Asplen |
| Pupal Degree Days - mean | 93.22 | Asplen |
| Pupal Degree Days - SD | 6.18 | Asplen |
| Adult Degree Days - mean | 1,050 | Asplen |

*Appendix 1—table 1 Continued on next page*

*Appendix 1—table 1 Continued*

| Parameter | Value | Reference |
|---|---|---|
| Adult Degree Days - SD | 40.41 | Asplen |

**Appendix 1—table 2.** Key to genotype nomenclature in the model.

| Allele | Definition |
|---|---|
| b | wildtype promoter for EGI target locus |
| B | resistant promoter for EGI target locus |
| d | wildtype allele at location of PTA integration |
| D | PTA targetting wildtype promoter b |
| p | wildtype promoter for EGI target locus |
| P | resistant promoter for EGI target locus |
| t | wildtype promoter for EGI target locus |
| T | PTA targetting wildtype promoter p |
| l | wildtype locus at the site of Female Lethal transgene integration |
| L | Female Lethality transgenic cassette |
| W | wildtype allele at locus of Gene Drive integration |
| G | Homing gene-drive allele |
| R | Gene Drive resistance allele with functional target locus |
| I | Gene Drive resistance allele that inactivates target locus |
| X | female sex chromosome |
| Y | male sex chromosome |
| q | promoter conversion of p (which behaves like P but is tracked independently) |
| r | natural resistance mutation in p that provides protection to T |
| z | promoter conversion of b (which behaves like B but is tracked independently) |
| c | natural resistance mutation in b that provides protection to D |

**Appendix 1—table 3.** Genotypes of wildtype or biocontrol agents released in simulations described in this study.

| Genotype in model | Definition |
|---|---|
| bbddppttllWWXX | wildtype female |
| bbddppttllWWXY | wildtype male |
| bbddPPTTllWWXX | EGI female |
| bbddPPTTllWWXY | EGI male |
| bbddppttLLWWXX | female-lethal female |
| bbddppttLLWWXY | female-lethal male |
| bbddppttLLWWXX* | RIDL female |
| bbddppttLLWWXY* | RIDL male |
| bbddPPTTLLWWXX | SSIMS female |
| bbddPPTTLLWWXY | SSIMS male |

*Removed "and self.sex=='female'" from line 236

