## [Editor Report]

This paper is of interest to entomologists caring for genetic pest control or molecular biologists following synthetic biology. The authors describe a fruit fly strain that combines constructs that establish both repressible female-lethality and genetic incompatibility based on CRISPR transactivation. They show that this strain has high penetrance for these two traits and that it can suppress wild-type flies when released into cycling cage populations.

---

## [Decision Letter]

**Decision letter after peer review:**

Thank you for submitting your article "Genetically engineered insects with sex-selection and genetic incompatibility enable population suppression" for consideration by *eLife*. Your article has been reviewed by 2 peer reviewers, and the evaluation has been overseen by a Reviewing Editor and Patricia Wittkopp as the Senior Editor. The following individual involved in review of your submission has agreed to reveal their identity: Ying Yan (Reviewer #2).

Essential revisions:

Both reviewers as well as the reviewing editor agree that this paper provides an interesting proof-of-concept for a genetic control strategy potentially applicable to insect pests and disease vectors. The cage suppression experiment (with the caveats mentioned in the reviews) is a nice set of data but the steps towards it have some issues. This study should warrant publication in *eLife* if the following essential revisions can be addressed.

1) We would like to see a better assessment of both male competitiveness and female fecundity, as laid out in the recommendations below.

2) These parameters should then be plugged into the model the authors developed previously to get a sense of how the strategy would compare.

3) A more balanced discussion should be provided, especially regarding potential resistances that directly link to the developed constructs/strains.

*Reviewer #1 (Recommendations for the authors):*

The "mating competition" experiment featured an excess of females rather than an excess of males. It is thus more a "mating capability" experiment as there is less need for males to compete for females. Can the authors explain why this was done like that and how it connects to the literature? Given what this paper tries to achieve, I would have liked to see a true competition experiment.

I'm missing the female fecundity data of the SSIMS strain. It is mentioned that the "slight sex-ratio biases […] might slow the expansion of a colony during mass rearing efforts" but female fecundity is also a major factor here. It can possibly extracted from the data underlying Figures 2b and c but it is far from clear.

Figure 3b. How was it determined that the females were non-virgin? Is this based simply on the expectation that they should all have mated after 24 hours?

*Reviewer #2 (Recommendations for the authors):*

Line 23: change "laboratory" into "mass". The scale of field SIT program is way beyond laboratory level.

Line 26: Cochliomyia hominivorax was also eradicated from the Central American until 2006. Please find the proper reference for that and I think it's better to give a whole picture here.

Line 32: delete "(cite Lance 2000)".

Figure 1c: please add copy number to tetO in the image; the blue and red box are not explained in the legend, and I suppose that they are the first exon and M exon in the Chtra gene, respectively, which are not applicable here since the tTA instead of Chtra is used.

A few words about the alternative splicing are necessary (either in the legend or text) as this is the key to female lethality. Currently it's not straightforward in the figure if male or female-specific splicing is designed in the construct, therefore I would delete the male intron in the diagram which is not used.

Line 42: the cited paper (Scott et al., 2020) is mainly genomic analyses, and it's "Concha, 2020, BMC Genetics" that actually did the work "Integrating an early acting female lethality system to New World Screwworm" as described in the text.

Line 45-46: this sentence gave impression that pgSIT insects were actually released into environment, which is not the case. Please consider to rephrase it.

Line 62: none of the cited papers developed the "the tetracycline (Tet)-repressible

positive feedback circuit". Instead, the authors may want to cite "Fu, 2007, Nat. Biotechnol." which is the first demonstration in insects and "Li, 2014, Insect Biochem and Mol Biol" for the development in *D. melanogaster*.

Line 68-79: although this study is a follow-up of previous reports from the same group, it should be able to stand on its own. Therefore, some more explanation about the previously characterized strains or tools are needed for general readers. For example, what's the rational for constructing two FL constructs into X chromosome (since most transgenic sexing strains of different insect species used one FL construct)? What does pyramus gene do? And "foxo promoter… performed especially well" in terms of what?

Line 92-94: this sentence is confusing: based on the figure 2c I would guess it's SSIMA X WT compared to WT X WT, but in the text it firstly mentioned "SSIMS males" then again "when mated with SSIMS males", so why SSIMA X WT compared to SSIMA X WT? In addition, the statistics suggested the difference is significant (p=0.02) so "a slight increase" is not a proper description. Also, for figure 2c it would be better to switch total offspring (to the right position) and egg-lay (to the left position as the 1st bar), since the eggs were produced/counted first before the adult offspring.

Figure 2: (b) shows 10 ;;g/ml Tet was not enough to supress the female lethality, then the authors increased Tet concentration all the way to 100 ;;g/ml "to further optimize rearing of SSIMS stock". But no statistics or conclusion was provided to show if these concentrations made difference. In other words, statistics should be included in (d) and (e) or result section.

Figure 2 legend: "(e) Percent female offspring when respective progeny from (c) are reared in food without Tet", (c) produce either WT progeny or no progeny so it can't be right here. More likely the (c) should be (d) and similarly the (d) afterwards should be (e). To make it clearer, better add the y and x axis title to each figure and use "F0, F1,.." to indicate the exact generation for certain Tet treatment or number scoring.

Line 105-113: the male mating competitiveness assay is problematic as it used 20 virgin females and 4 males, resulting 5:1 sex ratio so the males are not really competing for females. In addition, the controls with 1,2,3 WT males were directly compared to the mix tests using the same number of WT male, but they are not really comparable since the sex ratio is different (so the chance of encounter and other factors that affecting mating will be different in a given time and space). A more meaningful comparison/statistical analysis should be among the mix-males groups (since they have the same sex ratio) which I don't see. Nevertheless, I wouldn't call this mating competitiveness assay at all when a single male paired with five virgin females. This should be repeat with a more rational design (such as WT females: WT males: SSIMA males at 1:1:1 ratio).

Line 114-127: the sperm competition is usually carried out in two directions (the offense and defence assays) and the duration of the second mating is usually limited to a few hours to avoid multiple mating (Clark, 1995, Genetics; Yeh, 2012, PNAS; Jayaswal, 2018, Evolution; Lüpold, 2020, Evolution letter). This study performed offense assay but not defence assay, which can show the ability of SSIMA sperm to resist being displaced by subsequent sperm. Additional defence assay would provide a better assessment about the competitiveness of SSIMA sperm. While this could be optional, what is mind-troubling is that the mated females were housed with SSIMA males for a total of 12 days! Such period allows the females to mate with same or different males several times. In addition, it is known that seminal-fluid proteins mediate proper sperm storage and fertilization (such as the sex peptide binds to sperm) and the related network proteins quick degrade post mating. Figure 3 shows 5-8 days old females (in the text it says 5-6 days) were used, but it didn't specify the duration of the first crossing and importantly when the WT females were separated from the WT males before the second crossing. If those females were sired by WT males multiple days before the competition test, the long-stored WT sperm may be naturally become less competitive than SSIMA sperm which are freshly produced with seminal-fluid proteins. Therefore, it's necessary to give such critical information. Anyway, the lack of defence test and the long duration of the second crossing make this a sperm displacement assay rather than sperm competition assay. The authors should either repeat it with a strict time control, or soften their statements (in my opinion no conclusion can be made for sperm competitiveness).

Line 128-137: the female remating of D. suzukii has been reported from different groups (Revadi, 2015, Insects; Krüger, 2018, J. Appl. Entomol; Clark, 2020, J Insect Behav; Lanouette, 2020, The Canadian Entomologist), and almost all of them showed the low remating rate of this species. The authors should discuss and compare the high remating rate here to the previous studies. Again, the exact experimental set ups such as duration of and lag between the first and second crossing could play huge role in such test.

Line 186-191: this whole paragraph "…..This makes SSIMS one of the most complex engineered systems in insects" is somewhat contradictory to claim "the genetic design is likely to be portable to other species for applications in pest control" afterwards. With "23 synthetic genetic elements (operators, promoters, CDSs, terminators, etc.) spread across four chromosomal loci in the haploid genome", such design would be challenge to be transferred to other *Drosophila* species, and unimaginable to be transferred to species if either the genome or site-specific integration is not available. In addition, the design with so many functional elements also make it more vulnerable to the spontaneous mutation (Zhao, 2020, Nat Commu).

Line 202 and other places: is the SSIMS really capable of complete penetrance? The Figure 2 and 3 shows the offspring count for a few hundred and Figure 4 counts up to a thousand. Such number can hardly be used to predict field or mass-rearing condition. It has been reported that F1 survivors of a *Drosophila* Tetracycline-controlled genetic lethal strain (so immediately after release) could at a one out of 10,000 frequency due to known mutation in construct and even unknown suppressors that inherited maternally (Zhao, 2020, Nat Commu), and current New World Screwworm SIT facility in Panama release 15 millions sterile flies per week. What is even more critical, is that the actual basis for tTA lethality (original tTA overexpression system that adopted in this study) is still unknown which is subject to suppression by a pre-existing inherent variation in the targeted field population (Knudsen, 2020, G3). The very phenomenon may also be true for any gene-overexpression-based lethality including EGI lines generated here. The authors should discuss such potential resistances that directly link to their constructs.

Line 218-223: male-only was compared to bi-sex SSIMS release as "require larger release numbers to achieve suppression" but no data or modeling to support this, so the comparison is not grounded nor necessary here.

Line 228-229: this is cryptic and should be re-written.

Line 266: this is more like "lethality assay" rather than "mating assay". Where is the Tet treatments between 10 – 100 ;;g/ml?

---

## [Author Response]

Essential revisions:Both reviewers as well as the reviewing editor agree that this paper provides an interesting proof-of-concept for a genetic control strategy potentially applicable to insect pests and disease vectors. The cage suppression experiment (with the caveats mentioned in the reviews) is a nice set of data but the steps towards it have some issues. This study should warrant publication in eLife if the following essential revisions can be addressed.1) We would like to see a better assessment of both male competitiveness and female fecundity, as laid out in the recommendations below.2) These parameters should then be plugged into the model the authors developed previously to get a sense of how the strategy would compare.3) A more balanced discussion should be provided, especially regarding potential resistances that directly link to the developed constructs/strains.

Since these summary-level revision requests are described in detail by individual reviewers, I refer the editor to those responses (below). We have rigorously responded to these suggestions in the revised manuscript.

We are grateful for the detailed criticism provided by both reviewers. We have carefully considered their critiques and have improved our study by following their suggestions.

Reviewer #1 (Recommendations for the authors):The "mating competition" experiment featured an excess of females rather than an excess of males. It is thus more a "mating capability" experiment as there is less need for males to compete for females. Can the authors explain why this was done like that and how it connects to the literature? Given what this paper tries to achieve, I would have liked to see a true competition experiment.

We performed a mating competition experiment with an excess (i.e. non-limiting) number of adult females because this type of experiment was performed in the literature for a genetic biocontrol approach that is in the same application class as what we are reporting in this paper (Kandul, et al., 2019. https://doi.org/10.1038/s41467-018-07964-7). In that paper, male mating competitiveness is assayed at a 2 male : 10 female mating experiment, and experiments were performed with 2 wildtype, 1:1 wildtype to pgSIT, and 2 pgSIT males. In our original draft, we doubled the numbers of males and females (4 males: 20 females) to allow us to assess competition at 1:3, 2:2, and 3:1 mixed-male ratios.

We disagree with the reviewer that this is merely a ‘mating capability’ experiment, as mating capability can be assessed without males of each type. With that said, the excess number of females de-emphasizes a potentially important kinetic aspect to mating competition, wherein we might miss a competiveness phenotype that arises from GE males taking longer to initiate copulation with wild-type females. For this reason, we have performed a new mating competitiveness experiment (1:1:1) and include the data in a revised Figure 3a. This new experiment follows the literature precedent recommended by Reviewer #2. We continue to report the excess-female data as well, as this is still a good comparator to existing literature on SIT-like genetic biocontrol approaches, as mentioned above.

I’m missing the female fecundity data of the SSIMS strain. It is mentioned that the “slight sex-ratio biases […] might slow the expansion of a colony during mass rearing efforts” but female fecundity is also a major factor here. It can possibly extracted from the data underlying Figures 2b and c but it is far from clear.

This reviewer is correct that the fecundity data can be seen in Figures 2b and 2c (SSIMS male and female + Tet in Figure 2b, compared to wildtype male and female minus Tet in Figure 2c). We opted to report those data in separate subpanels since they are reporting the efficacy of separate genetic constructs (female lethality and EGI). For a more straight-forward comparison of fecundity between wild-type and the SSIMS strain, we added a supplementary figure (Supplementary Note 2) with a head-to-head comparison (new experiment). There was no statistically significant difference in fecundity.

Figure 3b. How was it determined that the females were non-virgin? Is this based simply on the expectation that they should all have mated after 24 hours?

We did not empirically test for prior mating before moving females to new tubes containing WT or SSIMS males in original Figure 3b (currently Figure 3c in the revised version). We performed a negative control experiment in which females from this same pool of putative non-virgins were moved to vials lacking males. The observation of reproduction in each of these control vials supports our assumption that the females had mated during the first 24 hours. However, this is different than direct empirical observation, so we note this weakness in our revised manuscript.

Also, we complemented this experiment with a new offense/defense sperm displacement experiment, in which the prior mating of females was confirmed empirically prior to moving to vials with new males (Figure 3d in revised manuscript). The new experiment mirror sperm displacement assays reported previously in the literature, as described in our response to Reviewer 2’s points below.

Reviewer #2 (Recommendations for the authors):Line 23: change "laboratory" into "mass". The scale of field SIT program is way beyond laboratory level.

Fixed

Line 26: Cochliomyia hominivorax was also eradicated from the Central American until 2006. Please find the proper reference for that and I think it's better to give a whole picture here.

We have revised this sentence for accuracy. We have replaced the two citations with a separate comprehensive review from Scott MJ et al., from 2017.

Line 32: delete "(cite Lance 2000)".Fixed.Figure 1c: please add copy number to tetO in the image; the blue and red box are not explained in the legend, and I suppose that they are the first exon and M exon in the Chtra gene, respectively, which are not applicable here since the tTA instead of Chtra is used.A few words about the alternative splicing are necessary (either in the legend or text) as this is the key to female lethality. Currently it's not straightforward in the figure if male or female-specific splicing is designed in the construct, therefore I would delete the male intron in the diagram which is not used.

We have added a more detailed description of the genetic design of both the female lethal and EGI cassettes in the figure, legend, and/or main text. We also describe the alternative splicing in the revised text. We have retained the male intron in the diagram, because previously published papers using this genetic design have shown that the mechanism of alternative splicing involves unique 5’ splice sites.

Line 42: the cited paper (Scott et al., 2020) is mainly genomic analyses, and it's "Concha, 2020, BMC Genetics" that actually did the work "Integrating an early acting female lethality system to New World Screwworm" as described in the text.

Thanks for catching this. We have revised with the appropriate citation.

Line 45-46: this sentence gave impression that pgSIT insects were actually released into environment, which is not the case. Please consider to rephrase it.

Fixed.

Line 62: none of the cited papers developed the "the tetracycline (Tet)-repressiblepositive feedback circuit". Instead, the authors may want to cite "Fu, 2007, Nat. Biotechnol." which is the first demonstration in insects and "Li, 2014, Insect Biochem and Mol Biol" for the development in *D. melanogaster*.

Fixed.

Line 68-79: although this study is a follow-up of previous reports from the same group, it should be able to stand on its own. Therefore, some more explanation about the previously characterized strains or tools are needed for general readers. For example, what's the rational for constructing two FL constructs into X chromosome (since most transgenic sexing strains of different insect species used one FL construct)? What does pyramus gene do? And "foxo promoter… performed especially well" in terms of what?

We clarified the text in this section to indicate that the two-copy FL strain produced a stronger female-lethal phenotype in a previous study, and we have cited that study. We also state that pyramus is a developmental morphogen. We have clarified that the EGI design integrated into SSIMS was selected because it provided strong genetic incompatibility in a previous study. We also note that SSIMS is expected to work with other specific genetic designs for the FL and EGI components as well.

Line 92-94: this sentence is confusing: based on the figure 2c I would guess it's SSIMA X WT compared to WT X WT, but in the text it firstly mentioned "SSIMS males" then again "when mated with SSIMS males", so why SSIMA X WT compared to SSIMA X WT? In addition, the statistics suggested the difference is significant (p=0.02) so "a slight increase" is not a proper description. Also, for figure 2c it would be better to switch total offspring (to the righ position) and egg-lay (to the left position as the 1st bar), since the eggs were produced/counted first before the adult offspring.

We have rearranged the bars in the figure as requested and clarified the description of these results (i.e. that we are comparing WT(f) x WT (m) to WT(f) x SSIMS(m)). We did not remove the descriptor ‘slightly’, because we see this change as modest. We do not think that the statistical significance (p=.02) makes it improper to describe the change as slight.

Figure 2: (b) shows 10 ;;g/ml Tet was not enough to supress the female lethality, then the authors increased Tet concentration all the way to 100 ;;g/ml "to further optimize rearing of SSIMS stock". But no statistics or conclusion was provided to show if these concentrations made difference. In other words, statistics should be included in (d) and (e) or result section.

We did not intend to communicate that the concentration of Tet was optimized in propagate greater numbers of SSIMS flies, but that it could be optimized to control the persistence of the genotype in biocontrol scenarios. Specifically, rearing parents in 10 ug/ml leads to 100% female lethality (within our limit of detection) in the F1 generation. Rearing parents in 100 ug/ml leads to 35% female survival, which in a field release scenario would allow for a subsequent generation (F2) that would be male-only. Rearing the original parents at concentrations between 10 ug/ml and 100 ug/ml does not impact female survival in the F1 generation.

We performed students t-test to confirm that there is no statistically significant difference between tetracycline concentration and the percentage of female offspring in figure 2d. We fit data in 2e (concentration dependent female survival in F1 reared without Tet) to a rectangular hyperbola (i.e., Hill function), which explains 86% of the variance in the data (R-squared = 0.86). This corresponds to a t-value of 24.3 and a p-value < 0.00001.

Figure 2 legend: "(e) Percent female offspring when respective progeny from (c) are reared in food without Tet", (c) produce either WT progeny or no progeny so it can't be right here. More likely the (c) should be (d) and similarly the (d) afterwards should be (e). To make it clearer, better add the y and x axis title to each figure and use "F0, F1,.." to indicate the exact generation for certain Tet treatment or number scoring.

Fixed.

Line 105-113: the male mating competitiveness assay is problematic as it used 20 virgin females and 4 males, resulting 5:1 sex ratio so the males are not really competing for females. In addition, the controls with 1,2,3 WT males were directly compared to the mix tests using the same number of WT male, but they are not really comparable since the sex ratio is different (so the chance of encounter and other factors that affecting mating will be different in a given time and space). A more meaningful comparison/statistical analysis should be among the mix-males groups (since they have the same sex ratio) which I don't see. Nevertheless, I wouldn't call this mating competitiveness assay at all when a single male paired with five virgin females. This should be repeat with a more rational design (such as WT females: WT males: SSIMA males at 1:1:1 ratio).

As described in our response to Reviewer #1, the mating competition assay in which females are not limiting was performed based on literature precedent. We have elected to keep this dataset in the revised manuscript, as it allows comparison of the SSIMS system to previously published genetic biocontrol approaches (that used a similar mating assay). To address this reviewer’s concern, we added a conventional 1:1:1 male mating competition experiment and show that there is a statistically significant decrease in the amount of offspring compared to a control treatment (new Figure 3a).

Line 114-127: the sperm competition is usually carried out in two directions (the offense and defence assays) and the duration of the second mating is usually limited to a few hours to avoid multiple mating (Clark, 1995, Genetics; Yeh, 2012, PNAS; Jayaswal, 2018, Evolution; Lüpold, 2020, Evolution letter). This study performed offense assay but not defence assay, which can show the ability of SSIMA sperm to resist being displaced by subsequent sperm. Additional defence assay would provide a better assessment about the competitiveness of SSIMA sperm. While this could be optional, what is mind-troubling is that the mated females were housed with SSIMA males for a total of 12 days! Such period allows the females to mate with same or different males several times. In addition, it is known that seminal-fluid proteins mediate proper sperm storage and fertilization (such as the sex peptide binds to sperm) and the related network proteins quick degrade post mating. Figure 3 shows 5-8 days old females (in the text it says 5-6 days) were used, but it didn't specify the duration of the first crossing and importantly when the WT females were separated from the WT males before the second crossing. If those females were sired by WT males multiple days before the competition test, the long-stored WT sperm may be naturally become less competitive than SSIMA sperm which are freshly produced with seminal-fluid proteins. Therefore, it's necessary to give such critical information. Anyway, the lack of defence test and the long duration of the second crossing make this a sperm displacement assay rather than sperm competition assay. The authors should either repeat it with a strict time control, or soften their statements (in my opinion no conclusion can be made for sperm competitiveness).

We have changed the language in our paper to call this a sperm displacement assay. We have clarified the figure and text to state that the females were mated for 5-6 days before immediately transferring to a new vial. We have performed additional experiments that replicate as closely as possible the sperm displacement experiments described in the (Clark 1995) paper cited by Reviewer 2. We observed a statistically significant decrease in the number of offspring on both the offense and defense assays compared to wt-mating controls. In both cases, the number of viable offspring decreased to approximately 1/3 that of the control. There was no difference between the offense or defense assays. We have updated Figure 3 with these results. We elected to retain both the original and new experiment in the revised manuscript. Both experiments contain appropriate controls, sample sizes, and statistical analysis. We describe in the revised Results section the unique insights that both experiments provide.

Line 128-137: the female remating of D. suzukii has been reported from different groups (Revadi, 2015, Insects; Krüger, 2018, J. Appl. Entomol; Clark, 2020, J Insect Behav; Lanouette, 2020, The Canadian Entomologist), and almost all of them showed the low remating rate of this species. The authors should discuss and compare the high remating rate here to the previous studies. Again, the exact experimental set ups such as duration of and lag between the first and second crossing could play huge role in such test.

We are thankful for this reviewer alerting us to these relevant publications, as there are important lessons/comparisons to be made between those works and our manuscript. Revadi et al., (2015) observed instances of female remating but did not quantify it in that publication. Kruger et al., (2018) observed low rates of re-mating when fertile females were co-housed with fertile virgin males and observed for a three-hour period each day. The opportunity for re-mating was limited to this three hour period, which had previously been established as a preferred mating window. Clark et al., (2020) report a decline in female mating frequency post copulation, from 50% to 16%. This was also measured in a short window (1 hour long) on intermittent days after the observed copulation event. The objective of Lanouette et al., (2020) was to compare remating frequencies of females that mated with wild-type or sterilize males, and they report low remating frequencies between 5 and 10% for both groups. Like the previous studies, females that mated with the first male were kept in isolation, and then reintroduced to a single male for a short window of two hours for remating.

As Reviewer 2 mentions, the experimental set ups are different in important ways. The previous reports on female remating employed direct observation of females that were kept in isolation outside of relatively short windows of potential mating. Our experiment did not rely on direct observation, but on unambiguous genetic data to determine the fraction of offspring that result from a remating event(s). This allowed us to continuously co-house the previously mated females with new males. Because (i) we do not need to manipulate the female flies after the first mating event (e.g., move them into a petri dish for direct observation), (ii) we can measure remating events that occur 24 hours per day for several days, and (iii) our flies are kept in social communities throughout the duration of the experiment, we feel that our experimental set-up more closely represents the opportunities to remate that females would experience in nature.

We understand that this SWD experiment seemed like a non-sequitur given our focus on the model system in *D. melanogaster* for the rest of the paper. Our intention was to illustrate that putative targets of genetic biocontrol are polyandrous, which has implications on the outcomes of various control approaches. In other words, we were more interested in a binary assessment of whether SWD females remate, which all of the papers cited above demonstrate. We have revised our manuscript to make this point using the cited literature, and we move our assay to the Supplemental Information section to preserve the focus of the paper on our model system.

Line 186-191: this whole paragraph "…..This makes SSIMS one of the most complex engineered systems in insects" is somewhat contradictory to claim "the genetic design is likely to be portable to other species for applications in pest control" afterwards. With "23 synthetic genetic elements (operators, promoters, CDSs, terminators, etc.) spread across four chromosomal loci in the haploid genome", such design would be challenge to be transferred to other *Drosophila* species, and unimaginable to be transferred to species if either the genome or site-specific integration is not available. In addition, the design with so many functional elements also make it more vulnerable to the spontaneous mutation (Zhao, 2020, Nat Commu).

This is a good point and is something that we have been thinking about in depth. We will add to this paragraph the downsides to the complexity of this system. Discussion of vulnerability to spontaneous mutation is discussed alongside other resistance mechanisms later in the Discussion section.

Line 202 and other places: is the SSIMS really capable of complete penetrance? The Figure 2 and 3 shows the offspring count for a few hundred and Figure 4 counts up to a thousand. Such number can hardly be used to predict field or mass-rearing condition. It has been reported that F1 survivors of a *Drosophila* Tetracycline-controlled genetic lethal strain (so immediately after release) could at a one out of 10,000 frequency due to known mutation in construct and even unknown suppressors that inherited maternally (Zhao, 2020, Nat Commu), and current New World Screwworm SIT facility in Panama release 15 millions sterile flies per week. What is even more critical, is that the actual basis for tTA lethality (original tTA overexpression system that adopted in this study) is still unknown which is subject to suppression by a pre-existing inherent variation in the targeted field population (Knudsen, 2020, G3). The very phenomenon may also be true for any gene-overexpression-based lethality including EGI lines generated here. The authors should discuss such potential resistances that directly link to their constructs.

We have revised the text to state that we favored ‘strong penetrance’ instead of ‘complete penetrance’. The sentence in line 202 was only meant to highlight or prioritization of penetrance over genetic burden. We address the limits of our study (including the inability to observe low frequency mutations) in new paragraphs added near the end of the discussion.

Line 218-223: male-only was compared to bi-sex SSIMS release as "require larger release numbers to achieve suppression" but no data or modeling to support this, so the comparison is not grounded nor necessary here.

We have added agent-based modeling to the paper to support this claim.

Line 228-229: this is cryptic and should be re-written.

We have re-written this paragraph to improve the clarity.

Line 266: this is more like "lethality assay" rather than "mating assay". Where is the Tet treatments between 10 – 100 ;;g/ml?

We have changed to “Offspring lethality assay”.